# All-ferroelectric implementation of reservoir computing

Zhiwei Chen[1], Wenjie Li[1], Zhen Fan [1][✉], Shuai Dong[1], Yihong Chen[1], Minghui Qin[1], Min Zeng[1], Xubing Lu [1], Guofu Zhou[2], Xingsen Gao [1] & Jun-Ming Liu [1,3]

Reservoir computing (RC) offers efficient temporal information processing with low training cost. All-ferroelectric implementation of RC is appealing because it can fully exploit the merits of ferroelectric memristors (e.g., good controllability); however, this has been undemonstrated due to the challenge of developing ferroelectric memristors with distinctly different switching characteristics specific to the reservoir and readout network. Here, we experimentally demonstrate an all-ferroelectric RC system whose reservoir and readout network are implemented with volatile and nonvolatile ferroelectric diodes (FDs), respectively. The volatile and nonvolatile FDs are derived from the same Pt/BiFeO$_3$/SrRuO$_3$ structure via the manipulation of an imprint field ($E_{imp}$). It is shown that the volatile FD with $E_{imp}$ exhibits short-term memory and nonlinearity while the nonvolatile FD with negligible $E_{imp}$ displays long-term potentiation/depression, fulfilling the functional requirements of the reservoir and readout network, respectively. Hence, the all-ferroelectric RC system is competent for handling various temporal tasks. In particular, it achieves an ultralow normalized root mean square error of 0.017 in the Hénon map time-series prediction. Besides, both the volatile and nonvolatile FDs demonstrate long-term stability in ambient air, high endurance, and low power consumption, promising the all-ferroelectric RC system as a reliable and low-power neuromorphic hardware for temporal information processing.

Deep learning is progressing rapidly and plays an increasing role in industry and daily life. It mainly relies on two types of neural network algorithms: feedforward neural network (FNN) and recurrent neural network (RNN), which are adept at handling static spatial and dynamic temporal tasks, respectively. Reservoir computing (RC) is a simple yet efficient type of RNN well suited for processing temporal information[1–3]. An RC system typically consists of a reservoir that nonlinearly maps the time-varying inputs into a high-dimensional feature space, and a readout network that performs further processing through a linearly weighted summation of the reservoir outputs (see Fig. 1a)[3]. During training, only the readout network needs to be trained while the reservoir does not. The training cost can thus be significantly reduced, which represents the most outstanding advantage of RC over other RNNs.

Recently, emerging hardware-based RC systems have attracted great attention, not only because they have achieved prediction performance comparable to that of the software-based counterparts in many tasks (e.g., pattern classification[4,5], speech recognition[6–8], chaotic system forecasting[6,7,9], and others[10–12]), but also because of their boosted energy efficiency[6,13]. For the hardware implementation of an RC system, the constituent reservoir and readout network need to be implemented on memory devices with distinctly different switching

[1]Institute for Advanced Materials and Guangdong Provincial Key Laboratory of Optical Information Materials and Technology, South China Academy of Advanced Optoelectronics, South China Normal University, 510006 Guangzhou, China. [2]National Center for International Research on Green Optoelectronics, South China Normal University, 510006 Guangzhou, China. [3]Laboratory of Solid State Microstructures and Innovation Center of Advanced Microstructures, Nanjing University, 210093 Nanjing, China. ✉e-mail: fanzhen@m.scnu.edu.cn

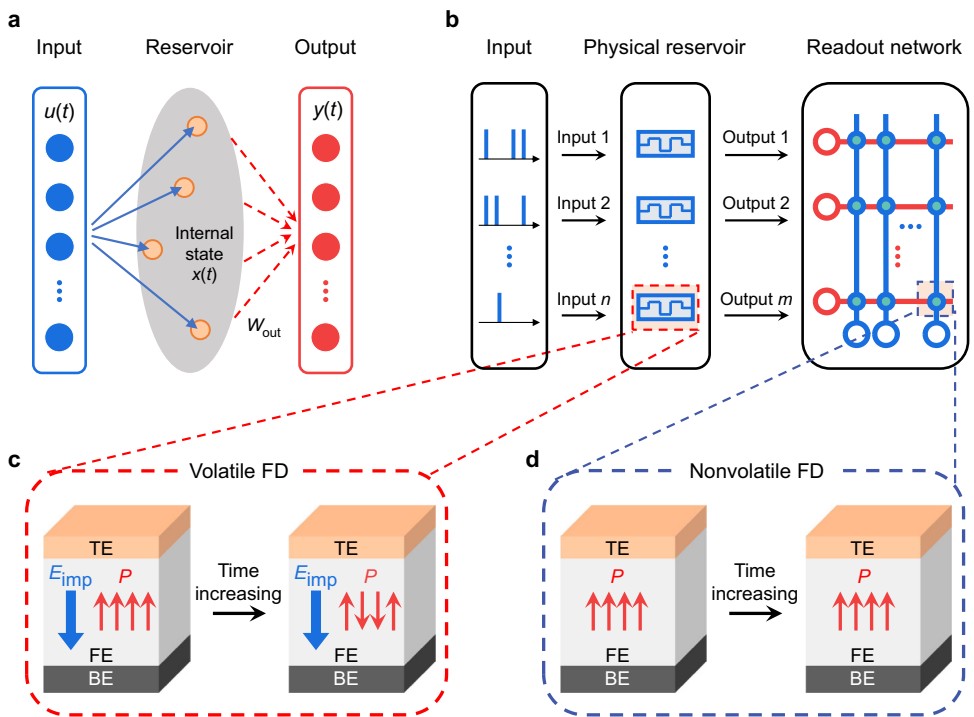

**Fig. 1 | Concept of all-ferroelectric RC system. a** Schematic of an RC system consisting of a reservoir with internal dynamics and a readout network. The inputs are projected into a high-dimensional feature space through the reservoir and then analyzed by the readout network. Only the weights in the readout network, i.e., $W_{out}$, need to be trained. **b** Schematic of an all-ferroelectric RC system, where the inputs are encoded as pulse trains while the reservoir and readout network are implemented with the volatile and nonvolatile FDs, respectively. Schematics of **c** volatile FD with $E_{imp}$ and **d** nonvolatile FD without $E_{imp}$. The $E_{imp}$ can cause polarization back-switching and consequent conductance decay. "TE", "FE" and "BE" denote the top electrode, ferroelectric, and bottom electrode, respectively.

characteristics, i.e., volatile and nonvolatile switching characteristics, respectively. Most previous studies have focused on the hardware implementation of the reservoir by using (volatile) diffusive memristors[6,7,10,14–22], nanomagnetic systems[23] (including spintronic oscillators[24], magnetic nanorings[25], spin ices[26], and magnonic systems[27]), self-organized nano-networks[9,28], electrochemical transistors[8,12,29], and so on. Among these devices, the diffusive memristors stand out because they possess intrinsic nonlinearity and short-term memory which are the two essential properties required by the reservoir[3], as well as high speed and excellent scalability. On the other hand, despite being less studied, the hardware implementation of the readout network has been demonstrated with (nonvolatile) drift memristors[9,13,21,30], whose nonvolatile conductances are utilized to map the weights in the readout network.

Notably, both the diffusive and drift memristors were mainly based on a filamentary mechanism, which can, however, lead to relatively large variations and low endurance due to the stochasticity of filament formation/rupture processes. This limits the prediction accuracy and reliability of the filamentary memristor-based RC system. Compared with filamentary switching, ferroelectric polarization switching is a more deterministic switching mechanism[31]. Ferroelectric memristors, which use polarization switching to tune the resistance[31,32], can thus exhibit highly reproducible memristive responses and potentially unlimited endurance[33–37]. Besides, they also show high switching speed and low-power consumption[38–40]. Using ferroelectric memristors as building blocks may therefore facilitate the development of highly reliable, accurate, fast, and energy-efficient ferroelectric-based RC systems.

However, the use of ferroelectric memristors in RC systems is currently scarce and mainly restricted to the reservoir[11,30,41–45], as summarized in Supplementary Table S1. All-ferroelectric implementation of a whole RC system still remains undemonstrated. The reason for this is probably because the ferroelectric memristors used

hitherto in RC systems—ferroelectric tunnel junction (FTJ)[30] and ferroelectric field-effect transistor (FeFET)[11,43,45]—possess inherently large depolarization fields ($E_{dp}$s) arising from ultra-small ferroelectric film thickness[46,47] and poor screening at ferroelectric/semiconductor interface[48], respectively. This makes them voluntary to exhibit volatile characteristics while difficult to be engineered into nonvolatile memristors to implement the readout network.

To construct an all-ferroelectric RC system, alternative ferroelectric memristors capable of being engineered into both volatile and nonvolatile memristors (for the reservoir and readout network, respectively) are demanded. A promising candidate is a ferroelectric diode (FD) which operates by using polarization to modulate the interfacial Schottky barrier[49–51]. FD is inherently subjected to a much smaller $E_{dp}$ compared with FTJ and FeFET, because it comprises a relatively thick ferroelectric film (several tens to hundreds of nanometers) sandwiched between two metal electrodes with good screening ability. Consequently, FD can readily function as a nonvolatile memristor[32,50,51]. In addition, FD can also be engineered to be volatile by judiciously introducing certain mechanisms for polarization back-switching[52–54]. Therefore, it is quite promising to use appropriately engineered FDs to implement both the reservoir and readout network, thus realizing an all-ferroelectric RC system in hardware (Fig. 1b–d).

In this work, we experimentally demonstrate an all-ferroelectric RC system consisting of a volatile FD-based reservoir and a nonvolatile FD-based readout network. The FDs with distinctly different volatile and nonvolatile switching characteristics are derived from the same capacitor-like structure of Pt/BiFeO$_3$ (BFO)/SrRuO$_3$ (SRO), which has not been realized yet for other types of ferroelectric memristors. The key to realizing this is purposely introducing an imprint field ($E_{imp}$) into the volatile FD while avoiding it in the nonvolatile FD. Owing to the $E_{imp}$, the volatile FD exhibits spontaneous polarization back-switching and consequent conductance decay, based on which short-term

memory and nonlinearity are further demonstrated. On the other hand, the nonvolatile FD with negligible $E_{imp}$ exhibits good polarization stability and consequent nonvolatile memristive switching, as well as long-term potentiation/depression (LTP/LTD). With these distinctly different device characteristics, the volatile and nonvolatile FDs are thus suitable building blocks of the reservoir and readout network, respectively. We then experimentally integrate them to build an all-ferroelectric RC system. Various tasks including curvature discrimination, digit recognition, waveform classification, and Hénon map prediction, are successfully implemented with the all-ferroelectric RC system, demonstrating competitive performance compared with the existing RC hardware systems. In particular, an ultralow normalized root mean square error (NRMSE) of 0.017 is achieved in the Hénon map prediction. Besides, long-term stability in ambient air, high endurance, and low-power consumption are proven in both the volatile and nonvolatile FDs, which can endow the all-ferroelectric RC system with high reliability and power efficiency. Our study showcases the development of application-specific neuromorphic devices based on ferroelectrics by manipulating the polarization dynamics and also highlights the great potential of ferroelectrics for use in hardware-based neuromorphic computing.

## Results

### Device structures

Two FDs with the same vertically stacked structure of Pt/BFO/SRO were fabricated, as schematically illustrated in Figs. 2a, 3a, respectively. All the deposition conditions for the BFO, SRO, and Pt films were kept the same, except that the BFO film in the nonvolatile FD was grown under an oxygen pressure of 15 Pa while that in the volatile FD was grown under 19 Pa (see the "Methods" section). Both the 15 and 19 Pa BFO films are phase-pure (Supplementary Fig. S1) and exhibit relatively flat surfaces (Supplementary Fig. S2). The ferroelectric and resistive switching properties of the two BFO-based FDs were investigated, where the voltages were applied to the Pt electrodes while the SRO electrodes were grounded.

### Electrical characteristics of nonvolatile FD

Figure 2b shows the ferroelectric polarization–voltage (P–V) hysteresis loop of the FD with the 15 Pa BFO film. A square P–V loop with almost symmetric coercive voltages (~±2.5 V) is observed, suggesting that negligible $E_{imp}$ exists in this device. The absence of $E_{imp}$ in this film is further evidenced by the piezoresponse force microscopy (PFM) results (Supplementary Fig. S3). Thanks to the absence of $E_{imp}$, both polarization up and down ($P_{up}$ and $P_{down}$, respectively) states are observed to be considerably stable (Supplementary Fig. S4). Such nonvolatile polarization may lead to nonvolatile memristive behavior given the polarization control of conduction in the FD, as demonstrated below.

Figure 2c shows the hysteretic current–voltage (I–V) characteristics of the FD with the 15 Pa BFO film. Typical switchable diode-type resistive switching behavior is observed. The critical voltages where the high resistance state (HRS) switches to the low resistance state (LRS) correspond well to the low-frequency coercive voltages (Supplementary Fig. S5), suggesting that the resistive switching is triggered by polarization switching. Such ferroelectric origin for the resistive switching can be attributed to the polarization modulation of interfacial Schottky barriers (Supplementary Figs. S6 and S7)[50,55].

The memristive behavior was further characterized using the pulse writing method. As shown in Fig. 2d, when applying a positive pulse train (amplitude: from 1.4 to 1.35 V in increments of 50 mV; width: 0.05 s) to the device, its conductance increases gradually from ~10 to ~36 nS. Moreover, all the conductance states (>4 bits) are considerably stable (Fig. 2e), which can be associated with the nonvolatility of polarization and polarization-controlled conduction in this device. Notably, the gradual increase of nonvolatile conductance well mimics the long-term potentiation (LTP) behavior of a bio-synapse. On the contrary, applying a negative pulse train (amplitude: −0.6 to −2.2 V in increments of 100 mV; width: 0.1 s) to the device results in long-term depression (LTD) behavior (Fig. 2d, f). Moreover, the LTP and LTD processes can be repeated for many cycles (Supplementary Fig. S8). Being capable of implementing the LTP and LTD functions, our FD with

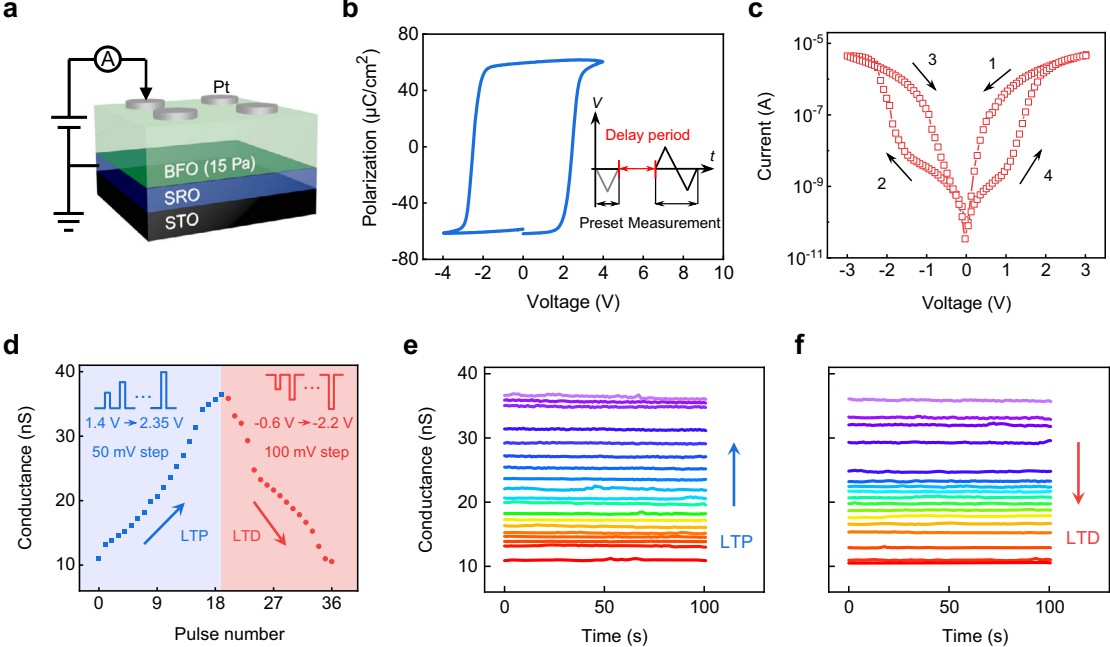

**Fig. 2 | Device characteristics of nonvolatile FD. a** Schematic of the Pt/BFO (grown under 15 Pa oxygen pressure)/SRO nonvolatile FD where $E_{imp}$ is absent. **b** P–V hysteresis loop measured using the applied waveform shown in the inset. The frequency of the measurement waveform is 3 kHz, and the delay period is 1 s (these parameters are always used for P–V loop measurements hereafter unless otherwise

mentioned). **c** I–V characteristics measured with a voltage sweep of 3 V → −3 V → 3 V. **d** LTP and LTD characteristics (read at 0.8 V) measured with amplitude-varying positive and negative pulses, respectively. The insets show the schematics of the applied positive and negative pulses. Retention behaviors of the conductance states obtained during the **e** LTP and **f** LTD processes.

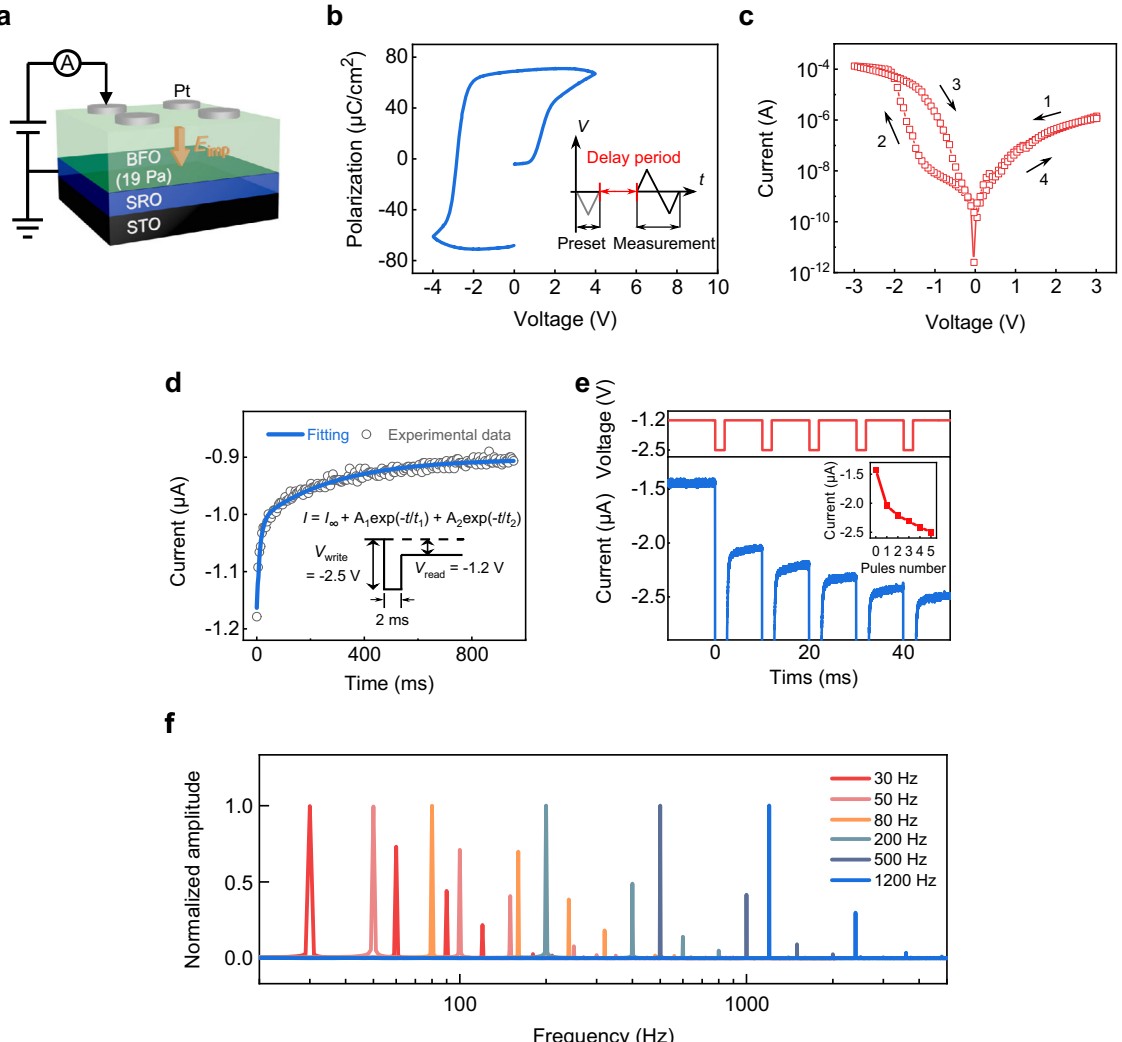

**Fig. 3 | Device characteristics of volatile FD. a** Schematic of the Pt/BFO (grown under 19 Pa oxygen pressure)/SRO volatile FD where $E_{imp}$ exists. **b** $P-V$ hysteresis loop measured using the applied waveform shown in the inset. **c** $I-V$ characteristics measured with a voltage sweep of 3 V → −3 V → 3 V. **d** Current decay characteristics after applying a −2.5 V/2 ms write pulse. **e** Current responses to 5 −2.5 V/2 ms write pulses. The upper panel shows the schematics of the applied write pulses followed by constant read voltages (−1.2 V). The inset shows the evolution of the read current as a function of the write pulse number. **f** FFT spectra of the output currents from the volatile FD which is subjected to input sinusoidal voltage waves with different frequencies.

the 15 Pa BFO film (hereafter termed the nonvolatile FD) can be used as the building block of the readout network (to be demonstrated later).

## Electrical characteristics of volatile FD

Compared with those of the nonvolatile FD, however, the ferroelectric and resistive switching behaviors of the FD with the 19 Pa BFO film become distinctly different. As shown in Fig. 3b, the $P-V$ loop of this device is shifted along the negative voltage axis, signifying the presence of a downward $E_{imp}$. In addition, there is a gap in the $P-V$ loop, which may originate from the back-switching of the upward polarization under the effect of the downward $E_{imp}$ during the delay period between the preset and measurement pulses (see inset in Fig. 3b). To further probe the polarization back-switching behavior, the polarization retention measurement was performed. As shown in Supplementary Fig. S9, the upward polarization decays over time while the downward polarization is considerably stable, pointing to the fact that the downward $E_{imp}$ exists and it induces the back-switching of only the upward polarization.

The finding of a downward $E_{imp}$ in the 19 Pa BFO film is further supported by the PFM results (Supplementary Fig. S3). Moreover, the $E_{imp}$ is observed to be stable against applied electric field and time at

room temperature (Supplementary Figs. S10–S12), and hence the $E_{imp}$-enabled functions (e.g., conductance volatility) are expected to work well. Prior to demonstrating this, it is necessary to understand why the 19 Pa film exhibits a downward $E_{imp}$ while the 15 Pa film does not. A depth-dependent X-ray photoelectron spectroscopy (XPS) study was therefore conducted. Supplementary Fig. S13 reveals that the oxygen vacancies are preferably distributed near the surface of the 19 Pa film, which may be the origin of the downward $E_{imp}$[51,52,56]. By contrast, it is revealed that although the amount of oxygen vacancies becomes larger in the 15 Pa film, they are relatively uniformly distributed throughout the film (Supplementary Fig. S13), accounting for the absence of $E_{imp}$.

The $E_{imp}$ and associated polarization back-switching have significant impacts on the resistive switching behavior, as demonstrated as follows. Figure 3c shows the hysteretic $I-V$ characteristics of the FD with the 19 Pa BFO film. The device exhibits one-side diode-type resistive switching behavior. Specifically, the HRS → LRS switching occurs only in the negative voltage region, which is associated with down-to-up polarization switching that lowers the height of the BFO/SRO barrier, i.e., the current-limiting barrier at negative voltages (see Supplementary Figs. S14–S16). However, negligible resistive switching

occurs in the positive voltage region, which may be due to two factors. First, some upward polarization already rotates in the downward direction before entering the positive voltage region because of the polarization back-switching, thus lowering the driving force for the resistive switching. Second, the Pt/BFO barrier, i.e., the current-limiting barrier at positive voltages, may be pinned at a high level because of the oxygen vacancies accumulated near the top interface (Supplementary Fig. S13), further suppressing the resistive switching in the positive voltage region (Supplementary Fig. S16)[51,52,57].

Besides the one-side diode-type resistive switching behavior, conductance volatility shall be another feature exhibited by the FD with the 19 Pa BFO film as a consequence of the $E_{imp}$ and associated polarization back-switching. To demonstrate it, the temporal conductance dynamics of the device were investigated. The device was first initialized to an intermediate state close to LRS by applying a $0 \rightarrow -3\,V \rightarrow 0$ sweep to it and subsequently leaving it unbiased for 3 min (hereafter unless otherwise specified, the device was always initialized to this intermediate state). Then, the device was stimulated with a write pulse (amplitude: $-2.5\,V$; width: $2\,ms$). Immediately after this, the current of the device was monitored with a read voltage stress of $-1.2\,V$. As shown in Fig. 3d, the read current decays over time, and the decay rate becomes slower with increasing time. The current decay can be well attributed to the polarization back-switching induced barrier height increase (Supplementary Fig. S17)[52]. The current decay curve can be further fitted to a double-exponential function (see inset in Fig. 3d), where two characteristic time constants $t_1 = 12\,ms$ and $t_2 = 280\,ms$, describing the fast and slow decay processes, respectively, are obtained.

Then, multiple write pulses with a short interval of 10 ms (shorter than $t_1$) were applied to the device. It is clearly seen from Fig. 3e that the read current after each write pulse is higher than that after its previous write pulse, mimicking the paired-pulse facilitation in a biosynapse[58,59]. However, when the interval between write pulses is sufficiently long (e.g., 1 s, which is longer than $t_2$), the read current does not increase with the number of write pulses anymore (Supplementary Fig. S18). The write pulse interval dependence of read current is well correlated with the competition between the history-dependent polarization switching driven by the write pulse and the polarization back-switching during the interval induced by $E_{imp}$. Combining the results of Fig. 3e and Supplementary Fig. S18, it is demonstrated that the device with the 19 Pa BFO film (hereafter termed as the volatile FD) exhibits short-term memory, i.e., its conductance state depends not only on the current input but also on the recent-past inputs.

In addition, the nonlinear $I-V$ characteristics (Fig. 3c) and the slowdown of the increasing rate of the read current with the write pulse number (inset in Fig. 3e) suggest the nonlinearity of our volatile FD. The nonlinear dynamics of the device can be further demonstrated through harmonic generation. Sinusoidal voltage waves with 6 different frequencies (30, 50, 80, 200, 500, and 1200 Hz) were sent to the device, and the output currents were recorded and are shown in Supplementary Fig. S19. Figure 3f shows the fast Fourier transform (FFT) spectra of the output current signals. For each input frequency, multiple higher harmonics are generated, validating the strong nonlinearity of the volatile FD.

Combining nonlinearity and short-term memory, the volatile FD can be exploited for the hardware implementation of the reservoir. Prior to demonstrating this, we note that both the nonlinearity and short-term memory of the volatile FD are associated with its complex polarization dynamics. The short-term memory is a result of both the history dependence of polarization switching[44] and the spontaneous polarization back-switching induced by $E_{imp}$. On the other hand, nonlinearity mainly originates from both the nonlinear polarization switching and the nonlinear polarization-controlled conduction behavior. Notably, polarization switching typically involves two microscopic processes: domain nucleation and domain growth, both of which have strong nonlinear dependencies on the applied voltage.

## All-ferroelectric RC system and its application in curvature discrimination

We can now construct an all-ferroelectric RC system by using the volatile and nonvolatile FDs as the building blocks of the reservoir and readout network, respectively. A simple task of curvature discrimination was first performed to demonstrate the capability of the all-ferroelectric RC system. 138 curves, half of which have positive curvature while the rest have negative curvature, are generated from a quadratic function (Supplementary Note 1). These curves are grouped into a training set (102 curves) and a test set (36 curves), as shown in Supplementary Fig. S20.

This task aims to determine whether the curvature of an input curve is positive or negative. As illustrated in Fig. 4a, each curve is first chopped into three sections, and each section is converted to a 3-timeframe pulse train. In each timeframe, the pulse amplitude is proportional to the relative height of the corresponding point on the curve, while the pulse width is fixed at 10 ms. The 3 pulse trains, as converted from one curve, are fed to a reservoir consisting of $N$ ($N = 3$ for this experiment) volatile FDs which work independently and in parallel, with each device processing one pulse train (see Fig. 4a). Different volatile FDs can exhibit different conductance states due to the different pulse histories, which is the key to extract the temporal information encoded in the pulse trains. The collective conductance states of all volatile FDs represent the reservoir state. To obtain the reservoir state, there are typically two approaches: (1) measuring the current responses of the volatile FDs to input write pulses and directly using them as the reservoir state[7] and (2) applying read pulses after input write pulses and using the read currents as the reservoir state[6,14]. The latter approach is used in this experiment.

The current signals output by the reservoir are converted to voltage signals, which are then fed to a readout network consisting of $M$ output neurons and $N \times M$ weights ($M = 1$ and $N \times M = 3$ for this experiment), as illustrated in Fig. 4a. The 3 weights and 1 bias are stored in 4 nonvolatile FDs (note: a pair of nonvolatile FDs can be used to represent a signed weight if needed). The readout network performs the dot product of the inputs (i.e., the reservoir outputs) and the weights connected to each output neuron, after which a sigmoid activation function is applied to generate the final neuronal output. The readout network is trained offline via logistic regression (see the "Methods" section), using the reservoir outputs from the training set as the inputs.

Figure 4b shows the photo of the experimentally constructed all-ferroelectric RC system for the curvature discrimination task. Figure 4c displays the reservoir states, as represented by the collective final conductance states of the 3-volatile FDs, after presenting 10 typical curves from the test set to the reservoir. The reservoir states corresponding to the positive and negative curvatures are clearly different.

These reservoir states, after being converted to voltage signals, are applied to the trained readout network. The resulting currents produced by the output neuron are shown in Fig. 4d and Supplementary Video 1. It is seen that the output currents corresponding to the negative curvature are apparently larger than those corresponding to the positive curvature. Feeding the output currents to the sigmoid activation function, the neuronal outputs are obtained and shown in Fig. 4e. One can observe that the neuronal outputs of the curves with negative curvature are all close to 1 while those of the curves with positive curvature are all close to 0. Besides these 10 typical curves, the rest curves in the test set are also correctly distinguished, giving rise to an overall accuracy of 100% (see Fig. 4f).

As comparisons, two control RC systems were designed and tested. In the first (second) control system, the volatile FDs in the reservoir are replaced with linear resistors (sigmoid functions), while the

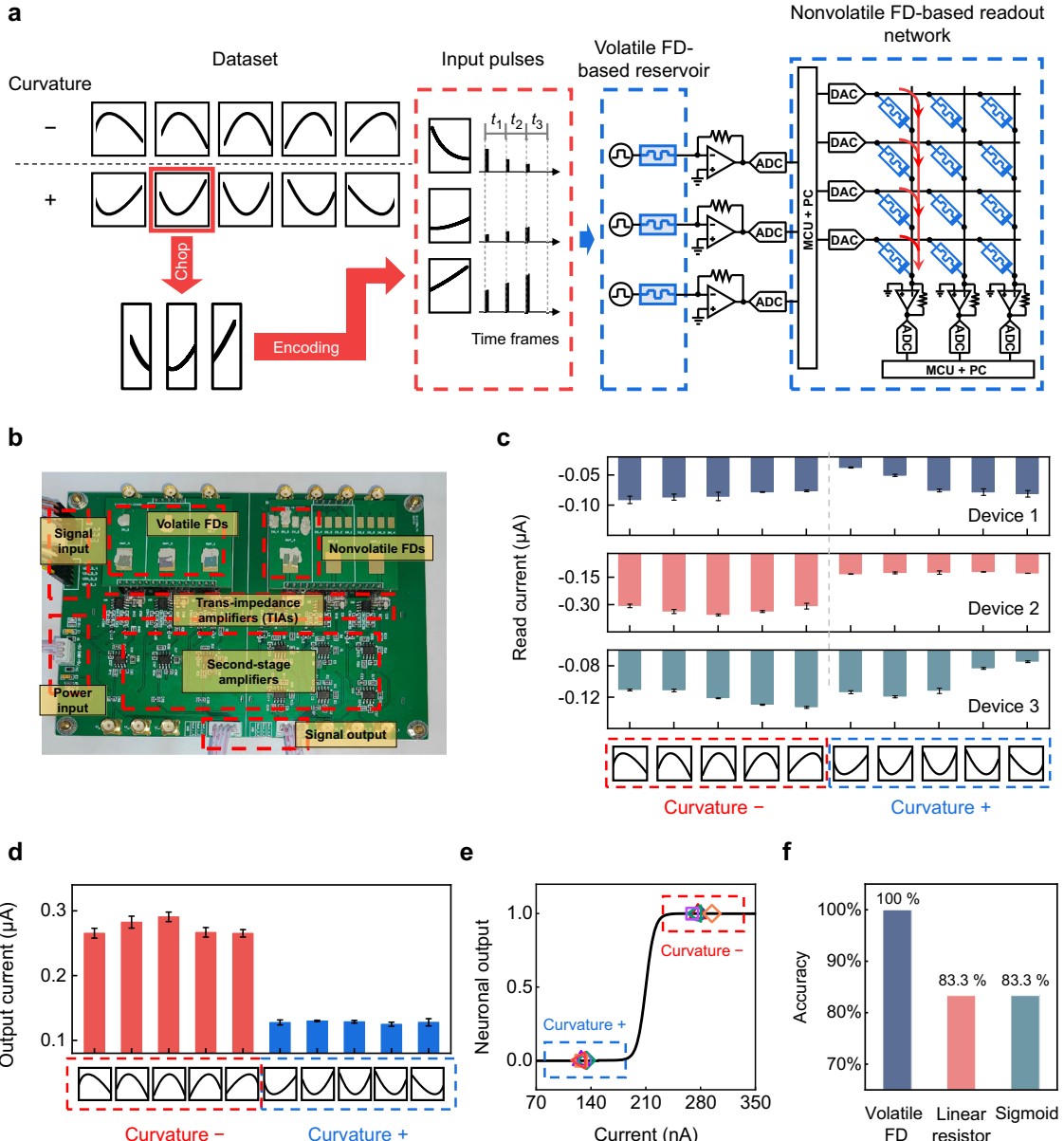

**Fig. 4 | Curvature discrimination. a** Schematic flow of the curvature discrimination task implemented on the all-ferroelectric RC system. Each input curve is chopped into 3 sections and then converted to 3 3-timeframe pulse trains, which are applied to a 3-volatile-FD reservoir. The reservoir states are subsequently fed to a nonvolatile FD-based readout network to obtain the final output. Note that only 1 column of nonvolatile FDs in the network is used in this task because it is a simple binary classification task. **b** Photo of the experimentally constructed all-ferroelectric RC system. **c** Experimentally measured reservoir states after presenting 10 typical curves from the test set to the 3-volatile-FD reservoir. **d** Output currents generated by the output neuron in the readout network during the presentations of the 10 typical curves. The error bars are obtained from 4 independent experiments. **e** Neuronal outputs after feeding the output currents in (**d**) to the sigmoid activation function. **f** Comparison of the accuracies on the test set (36 curves) between the all-ferroelectric RC system with a volatile FD-based reservoir and two control RC systems with linear resistor- and sigmoid-based reservoirs.

pre-/post-processing processes remain unchanged. As shown in Fig. 4f, both the linear resistor- and sigmoid-based RC systems achieve apparently lower accuracies than the all-ferroelectric RC system. This confirms that the volatile FD offering both nonlinearity and memory effect is the key to the reservoir, and the volatile FD-based reservoir is more critical than the pre-/post-processing[60] for our RC system's capability of curvature discrimination (see Supplementary Fig. S21 and Note 2 for more detailed discussion).

## Digit recognition

Then, a more complex task of 10-class digit recognition was carried out. Figure 5a shows the 10 digits represented by the 5 × 3

images used for training. As seen from Fig. 5b, the input image is first divided into 5 rows, and each row is converted to a 3-timeframe pulse train. In each timeframe, either a −2.5 V/2 ms write pulse (corresponding to the white pixel '1') or zero voltage (corresponding to the black pixel '0') is applied. In this way, the original spatial features of the image are now encoded as temporal information in the pulse trains. The 5 pulse trains are then fed to a reservoir consisting of 5-volatile-FDs in parallel.

For the 10-digit images used here (Fig. 5a), there are 7 different possible pixel arrangements along the row: "111", "110", "101", "100", "011", "010", and "001", corresponding to 7 different pulse trains. When the 7 different pulse trains are applied to a

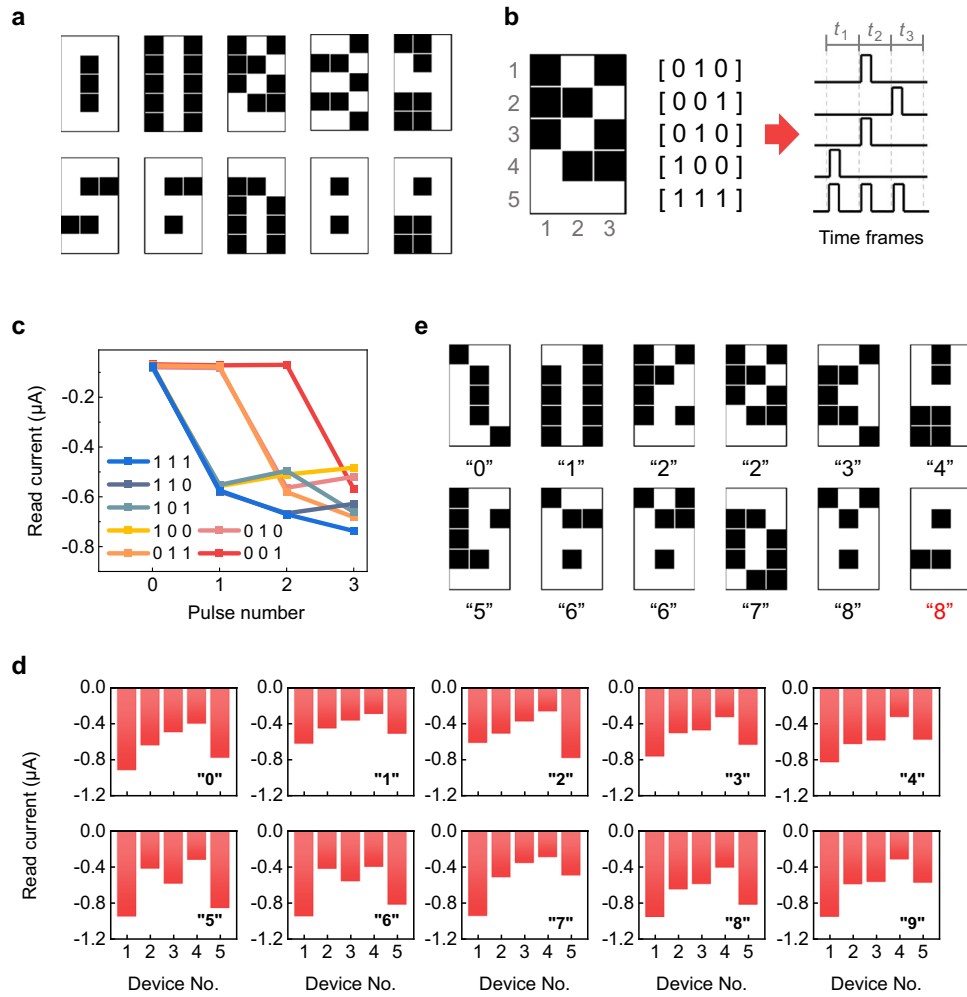

**Fig. 5 | 10-class digit recognition. a** Images of the 10 digits in the training set. **b** Example of the conversion from a 5 × 3 image to 5 3-timeframe pulse trains. **c** Read current evolutions of a typical volatile FD under 7 different pulse trains representing the pixel arrangements of "111", "110", "101", "100", "011", "010", and "001". **d** Experimentally measured reservoir states after stimulating the 5-volatile-FD reservoir with different input images in the training set. **e** Images of the 12 digits in the test set, which are the noisy versions of their corresponding training samples in (**a**). The noisy "9" is incorrectly recognized as "8" (marked in red), whereas all the rest digits are correctly recognized.

typical volatile FD in the reservoir, the corresponding conductance evolutions are shown in Fig. 5c. The other 4 FDs in the reservoir exhibit similar conductance evolutions, as demonstrated in Supplementary Fig. S22. Thanks to their nonlinearity and short-term memory, all the devices exhibit clearly separated final conductance states when subjected to the different pulse trains, indicating their capability to distinguish these input pulse trains. This in turn renders the reservoir capable of generating separable reservoir states when fed with different images. Figure 5d shows the reservoir states after the stimulations with different images, as represented by the combination of the final conductance states of all the 5 volatile FDs. These reservoir states are apparently different, demonstrating the good separation property of the volatile FD-based reservoir. However, the reservoir's separation property becomes much poorer after replacing the volatile FDs with linear resistors (Supplementary Fig. S23), verifying the critical role played by the volatile FDs.

The reservoir states produced by the volatile FD-based reservoir were subsequently used as inputs to the readout network. However, due to the relatively large size of the readout network (6 × 10; including biases), it was difficult to implement it with the nonvolatile FDs using the wiring method (see the "Methods" section). The readout network was therefore

simulated. Nevertheless, the software-computed floating-point weights were not directly used; instead, the weights were mapped onto the experimentally measured conductance values of nonvolatile FDs (Fig. 2d). Hereafter unless otherwise specified, the readout network was always simulated in this way, but the reservoir was still experimentally implemented with volatile FDs.

Supplementary Fig. S24 shows that 100% accuracy is achieved after fewer than 50 epochs of training, evidencing accurate and fast training. After training, 12 noisy digits (Fig. 5e), which were not included in the training set, were presented to the all-ferroelectric RC system for the test. The all-ferroelectric RC system correctly recognizes 11 out of 12 digits, realizing an accuracy of 91.7%. Only the noisy "9" is incorrectly recognized as "8", which is not unacceptable because there is only 1-pixel difference between the noisy "9" and the original version of "8" (see comparison between Fig. 5a, e). Note that the 91.7% accuracy achieved by the all-ferroelectric RC system on the test set is the same as that achieved by an ideal fully connected neural network (FCNN). However, the number of weights needing to be trained in the all-ferroelectric RC system is only 60 (6 × 10), while that increases to 160 (16 × 10) in the FCNN. This highlights the low training cost of the all-ferroelectric RC system.

Besides, the MNIST handwritten digit recognition was also implemented (Supplementary Figs. S25, S26). The all-ferroelectric RC

system achieves an accuracy of 89.5%, which is 6.5% higher than that of a pioneering diffusive memristor-based RC system[21].

## Waveform classification

For the above curvature discrimination and digit recognition tasks, the original input data are indeed not time-dependent and they are artificially converted to temporal data. More native applications of RC systems may be processing original temporal data directly. One representative example of such applications is waveform classification, which was implemented with our all-ferroelectric RC system. In this experiment, the input sequence is composed of randomly generated sine and square waveforms. Each data point in the input sequence is multiplied by a mask which is a 1D vector with a length of 6 composed of randomly assigned binary values of 0 and 1, and then converted to a 6-timeframe pulse train containing only write pulses with no read pulses (see Fig. 6a). The write pulse has an amplitude in the range of −2.3 to 0 V as linearly scaled from the masked input. Both the pulse width and interval are 3 ms, making the total length of the pulse train ($\tau$) equal 36 ms. As 8 masks are used here, one input data point can thus generate 8 pulse trains, which are, respectively, applied to the 8 parallel volatile FDs in the reservoir. For each volatile FD, 6 virtual nodes are created due to the application of the 6-timeframe pulse train, and the current responses to the write pulses are directly used as the virtual nodes' states. The combination of the virtual-node states of all volatile FDs forms the reservoir state, and hence the reservoir size is expanded from 8 to 48 (6 × 8). The reservoir state is then fed to a 49 × 1 (including a bias) readout network to perform the classification. The readout network is trained with linear regression, and the target output is a binary sequence with −1 and 1 representing sine and square waveforms, respectively.

As shown in Fig. 6b, the trained all-ferroelectric RC system can correctly classify the sine and square waveforms into their corresponding categories with an NRMSE of 0.13. This NRMSE value is sufficiently low, even lower than the value of 0.2 reported recently in an $\alpha$-In$_2$Se$_3$ FeFET-based RC system[11] which used high-precision floating-point weights for the readout network (note: in our work, the readout weights are mapped onto the measured conductances of nonvolatile FDs). Such low NRMSE of our all-ferroelectric RC system is attributed to the capability of the volatile FD-based reservoir to produce sufficiently high feedback strength and state richness (Supplementary Fig. S27).

## Hénon map prediction

To further evaluate the performance of our all-ferroelectric RC system on temporal signal processing, a benchmark task for time-series prediction, i.e., Hénon map prediction was demonstrated. The Hénon map is a typical discrete-time dynamic system exhibiting chaotic behavior[61]. It takes a point $(x(n), y(n))$ in the 2D plane and maps it to a new point $(x(n+1), y(n+1))$ through the equations below:

$$x(n+1) = y(n) - 1.4x(n)^2, \tag{1}$$

$$y(n+1) = 0.3x(n) + w(n), \tag{2}$$

where $w(n)$ is a Gaussian noise whose mean value and standard deviation are 0 and 0.05, respectively. The task is to predict $(x(n+1), y(n+1))$, given the $(x(n), y(n))$ values up to the time step $n$. Substituting Eq. (2) into Eq. (1) results in an equation containing only $x$, and hence the task becomes the prediction of $x(n+1)$ based on the $x(n)$ and $x(n-1)$ values.

When implementing this task with the all-ferroelectric RC system, a dataset of an $x(n)$ series with a length of 500 is generated through iterations with Eqs. (1), (2), where the first 300 data points are used for the training while the rest are used for the

test. An input $x(n)$ value is converted to pulse trains through a mask process similar to that used for the waveform classification. However, the mask length and the number of masks become 3 and 8, respectively. Correspondingly, each pulse train has 3 write pulses (amplitude: from −2 to 0 V as linearly scaled from the masked input; width: 2 ms, interval: 2 ms), and the number of pulse trains is 8. Applying these pulse trains to an 8-volatile-FD reservoir produces 24 virtual-node states (3 × 8) corresponding to an input $x(n)$ value. Similarly, an input $x(n-1)$ value can also generate 24 virtual-node states. The total 48 virtual-node states are combined and fed to a 49 × 1 readout network (including a bias) to predict the $x(n+1)$ value, and the readout network is trained with linear regression.

Figure 6c, e show the time series predicted by the all-ferroelectric RC system during the training and test processes, respectively, which agree with their corresponding ideal targets fairly well. Figure 6d, f, which are the 2D plots of the results in Fig. 6c, e, respectively, demonstrate that the strange attractor of the Hénon map can be well reconstructed. The NRMSE value on the test set is further calculated to be 0.017, which is an ultralow value among those reported for the RC hardware systems[6,7,9,62]. Such good performance can mainly be attributed to the reservoir's strong capability to capture temporal features (Supplementary Fig. S28), which stems from the complex polarization dynamics of volatile FDs. Other possible factors are analyzed in Supplementary Fig. S29 and Note 3.

## Device reliability and power consumption

Good reliability is expected for ferroelectric memristors due to the polarization control of conductance. For both our volatile and nonvolatile FDs, the cycle-to-cycle (C2C) variations are observed to be ≤-8% (Supplementary Fig. S30). In particular, the small C2C variation of the volatile FD is the key to the accurate temporal signal transformation through the reservoir. The device-to-device (D2D) variations of both the volatile and nonvolatile FDs are, however, relatively large (≤-60%; see Supplementary Fig. S31). Nevertheless, the D2D variation of the volatile FD is indeed favorable to expanding the reservoir size, which can help to improve the RC performance. On the other hand, the D2D variation of the nonvolatile FD is not a big issue because of the inherent fault tolerance of the readout network. In addition, applying a write-and-verify method to the nonvolatile FD can ensure relatively precise weight programming despite the existence of C2C and D2D variations. In fact, for the nonvolatile FD, good retention, as demonstrated in Fig. 2e,f, is of critical importance for the RC performance. Moreover, both volatile and nonvolatile FDs have at least 30-day stability in ambient air (see Supplementary Fig. S32), which is an advantage over some liquid electrolyte-based devices whose performance may degrade when exposed to ambient air for a long time[63]. In terms of endurance, both volatile and nonvolatile FDs can be switched for at least $10^6$ cycles with little polarization degradation and little resistance state drift (Supplementary Fig. S33). Last, but not least, the volatile FD consumes -11.8 µW per input while the nonvolatile FD consumes -140 nW per input (Supplementary Note 4). Such power consumptions are at least 3 times lower than those of the state-of-the-art filamentary memristors used for the RC hardware systems[6,7,16], suggesting the potentially high-energy efficiency of our all-ferroelectric RC system.

## Discussion

In summary, we experimentally demonstrate an all-ferroelectric RC system in which the reservoir and readout network are implemented with the volatile and nonvolatile FDs, respectively. Both the volatile and nonvolatile FDs have the same structure of Pt/BFO/SRO, but the difference is that $E_{imp}$ is purposely introduced into the former while it

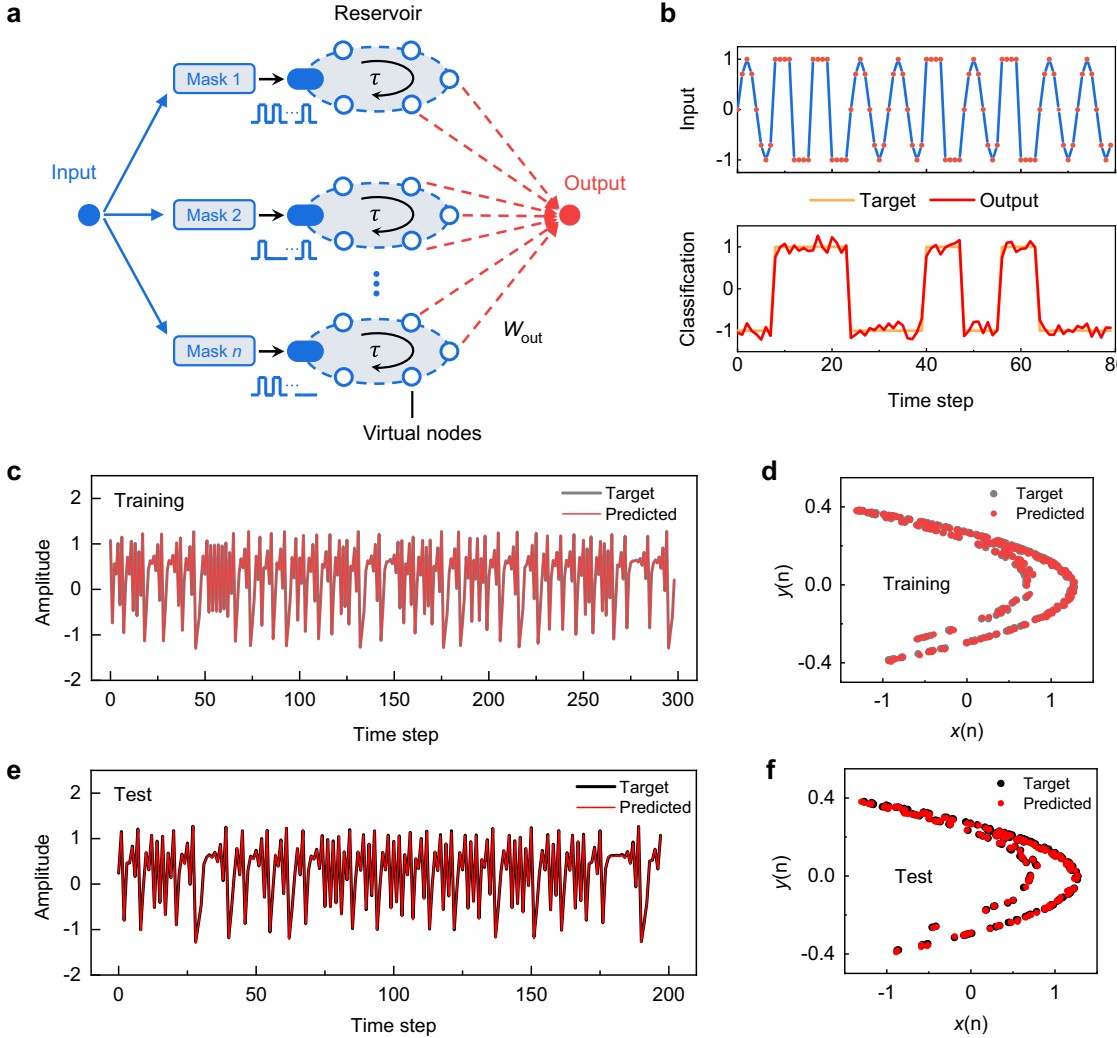

**Fig. 6 | Waveform classification and Hénon map prediction. a** Schematic illustration of the mask process used to generate virtual nodes. **b** Inputs of sine and square waveforms (upper panel) and their classification results obtained from the all-ferroelectric RC system (lower panel). **c, e** Predicted time series versus ideal targets and **d, f** corresponding 2D plots on the **c, d** training and **e, f** test sets. The predicted results in (**c**–**f**) are obtained from the all-ferroelectric RC system.

is absent in the latter. Under the effect of $E_{imp}$, the volatile FD displays spontaneous polarization back-switching and consequent conductance decay. Short-term memory and nonlinearity are further demonstrated in the volatile FD. These properties enable the volatile FD to produce well-separable responses to different temporal inputs, making it a suitable building block of the reservoir. On the other hand, the nonvolatile FD with negligible $E_{imp}$ exhibits good polarization stability and consequent nonvolatile memristive switching. Moreover, the LTP and LTD functions are implemented in the nonvolatile FD, qualifying it as the synapse in the readout network. Then, an all-ferroelectric RC system consisting of the volatile FD-based reservoir and the nonvolatile FD-based readout network is constructed. The all-ferroelectric RC system is used to solve various tasks including curvature discrimination, digit recognition, waveform classification, and Hénon map prediction, and it achieves competitive performance compared with the existing RC hardware systems. In particular, an ultralow NRMSE of 0.017 is achieved in the Hénon map prediction. Besides, both volatile and nonvolatile FDs demonstrate long-term stability in ambient air, high endurance, and low power consumption, making the all-ferroelectric RC system a reliable and low-power hardware platform for temporal information processing. We expect that our encouraging results will stimulate further research on the ferroelectric implementation of various emerging neuromorphic computing algorithms, e.g., e-prop[64].

## Methods

### Device fabrication

BFO epitaxial thin films (~130 nm) were grown on (001)-oriented SrTiO₃ substrates with SRO bottom electrode layers (~40 nm) by pulsed laser deposition using a KrF excimer laser ($\lambda = 248$ nm). Laser energy fluences used to deposit SRO and BFO films were ~1.0 and ~1.1 J/cm², respectively, while the repetition rates used for both films were the same, i.e., 5 Hz. The SRO films were first deposited on the STO substrates, during which the substrate temperature and oxygen pressure were kept at 680 °C and 15 Pa, respectively. The BFO films were subsequently deposited at an elevated substrate temperature of 690 °C under different oxygen pressures (19 and 15 Pa for volatile and nonvolatile FDs, respectively). After growth, the samples were cooled to room temperature at a 10 °C/min rate in an oxygen atmosphere of 1000 Pa. Then, circular Pt top electrodes (~100 μm in diameter) were deposited on the BFO/SRO films by PLD at room temperature through a shadow mask. The individual Pt/BFO/SRO FDs, including both volatile FDs and nonvolatile FDs, were thus obtained. To construct an all-ferroelectric RC system, the volatile and nonvolatile FDs were

mounted on a test board and wired to their corresponding input and output pads using Pt wires and silver pastes.

## Structural, morphological, and elemental characterizations

The crystalline structures of the BFO films were examined by X-ray diffraction (XRD) using a PANalytical X'Pert PRO diffractometer. The morphologies and domains were characterized by atomic force microscopy (AFM) and piezoresponse force microscopy (PFM), respectively, using an Asylum Research MFP-3D AFM system and Pt-coated silicon tips (Nanoworld EFM Arrow). X-ray photoelectron spectroscopy (XPS) was conducted for elemental analysis using a Thermo Fisher Scientific ESCALAB 250Xi system with Al Kα source (1486.6 eV). To allow the depth-dependent XPS study, $Ar^+$ ion etching was performed on the samples.

## Electrical measurements

All electrical measurements were carried out on a custom-built probe station in the air. $P–V$ hysteresis loops were measured with a ferroelectric workstation (Radiant Precision Multiferroic). DC $I–V$ measurements were conducted with a Keithley 6430 SourceMeter. In the pulse measurements, an Agilent 33250A function generator was used to generate voltage pulses while the resultant currents were recorded using a combination of a LeCory 64Xi-A oscilloscope and amplifier circuits. Electrical measurements on the RC system were realized with the test board (containing volatile and nonvolatile FDs and peripheral circuits), a microcontroller unit (MCU), an 8-channel 16-bit analog-to-digital converter (ADC), 12-bit digital-to-analog converters (DACs), and a personal computer (PC). The peripheral circuits on the test board consisted of trans-impedance amplifiers (TIAs) and second-stage amplifiers, which were used to first convert currents to voltages and then amplify the voltage signals. The MCU used in our experiment was STM32, which was responsible for generating and reading voltage signals through DACs and ADCs, respectively. The PC was used to run the basic loop of the RC algorithm coded in Python, and it communicated with STM32 via a universal asynchronous receiver/transmitter (UART). In the curvature discrimination task, a user interface was coded in Python to load, process, and save the data, and display the classification results.

## Simulations

For the curvature discrimination task, a sigmoid activation function was used to calculate the output categorical probability, which is expressed as

$$\hat{y}_i = \text{sigmoid}(x_i) = \frac{1}{1 + e^{-x_i}}, \tag{3}$$

where $x$ is the neuronal input scaled from the measured current $I$:

$$x = \alpha(I - \beta), \tag{4}$$

where $\alpha$ is a scaling factor and $\beta$ is an offset ($\alpha = 0.18\,nA^{-1}$ and $\beta = 210\,nA$ in this work). A different sigmoid function was used to replace the volatile FD to construct a control RC system, and its expression is presented in the Supplementary Information.

In addition, the readout network was trained via the regularized logistic regression and the cost function ($J$) is expressed as

$$J = \frac{1}{m} \sum_{i=1}^{m} \left[ -y_i \log(\hat{y}_i) - (1 - y_i) \log(1 - \hat{y}_i) \right] + \lambda \sum_{j=1}^{n} |w_j|, \tag{5}$$

where $m$ is the number of samples, $y_i$ is the desired target, $n$ represents the number of weights, and $\lambda$ is the regularization parameter.

For the digit recognition task, a supervised learning algorithm, i.e., softmax regression, was used to fit the weights of the readout layer. A softmax function was used as the activation function of the readout network to calculate the probabilities corresponding to the different possible outputs, which are expressed as

$$\hat{y}_i = \text{softmax}(z_i) = \frac{e^{z_i}}{\sum_{j=1}^{N} e^{z_j}}, \tag{6}$$

where $z_i$ is the original neuronal output, $\hat{y}_i$ is the probability corresponding to $z_i$, and $N$ is the total number of output neurons, i.e., the number of categories of classification. In the training process, the categorical cross-entropy loss function ($L$) was used, which is given by

$$L = -\sum_{i=1}^{m} y_i \log(\hat{y}_i). \tag{7}$$

In the waveform classification and Hénon map prediction tasks, the linear regression was used to train the readout network with a least-squares method, which is expressed as

$$\mathbf{W} = (\mathbf{X}^T \mathbf{X})^{-1} \mathbf{X}^T \mathbf{Y}, \tag{8}$$

where $\mathbf{W}$ is the weight matrix of the readout network, $\mathbf{X}$ is the input matrix, and $\mathbf{Y}$ is the target matrix.

## Data availability

The data that support the findings of this study are available from the corresponding author upon reasonable request.

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

## Acknowledgements

The authors would like to thank the Science and Technology Projects in Guangzhou (202201000008), the National Natural Science Foundation

of China (Nos. 92163210, U1932125, 52172143), and the Science and Technology Program of GuangZhou (No. 2019050001).

## Author contributions

Z.F. conceived and supervised the research. Z.C. and W.L. prepared the devices. Z.C., W.L., Z.F., M.Z., and X.L. performed the electrical measurements. Z.C., W.L., G.Z., and X.G. conducted the XRD, AFM, and XPS measurements. Z.C., S.D., Y.C., and M.Q. carried out the simulation works. Z.C., Z.F., X.G., and J.-M.L. wrote and revised the manuscript.

## Competing interests

The authors declare no competing interests.
