## [Peer Review File · Nature Communications]

REVIEWER COMMENTS

Reviewer #1 (Remarks to the Author):

Nature Comms review questions – all-ferroelectric reservoir computing

Initial Comments

The authors present a thorough study on an all ferroelectric memristor computation system, split roughly into two parts: description of the material synthesis and device architecture and discussion of some benchmark reservoir computation tasks. The work seems solid if a little difficult to understand at times due to its long, dense nature, formatting issues and lack of clarity on some key points (see below). Overall it is a nice work, some important clarifications need to be made before it is possible to recommend it for publication in Nature Communications.

I am not a ferroelectric scientist by training, and while the material system results certainly look nice, thorough and well-presented – I lack the grounding in the literature to assess whether there is strong novelty in the design or implementation of the ferroelectric memristors used here. The authors could make a better attempt at situating this section within the literature for the broad readership at the multidisciplinary journal they are submitting to.

My expertise here is on the neuromorphic computing side, and it must be said that the Henon map results are very impressive, the authors have done a commendable job at this challenging task. However, it is not explained why the Henon map performance is so good, especially when the other computing results are not particularly impressive and are far from cutting edge. Is it because of some new physics on show here? as the reservoir architecture is fairly standard. Unfortunately this is not really commented on, so it's hard to know whether it just so happens that the dynamics of the system here are a good match for the Henon attractor (this is a known phenomena in neuromorphic computing, each chosen physical system will be better or worse suited to given nonlinear problems) or if there is some exceptional computing going on.

In trying to ascertain this, the reader's job is complicated by a lack of control experiments to indicate how much of the observed computing performance is achieved via pre-processing, virtual nodes and offline nonlinear post-processing, and how much is achieved via the unique qualities of the ferroelectric system. For me, gaining understanding of this is the deciding factor on whether this work is fit for Nature Communications or if it would be a better fit for Communications Physics/Communications Engineering.

The concept is not novel – memristive reservoir computing has been around for some time, as has ferroelectric reservoir computing. I would like more concrete comparisons to other ferroelectric computing systems specifically to ascertain where the novelty lies and how novel and important these results are. Particularly these recent works:

“Multilayer Reservoir Computing Based on Ferroelectric α -In₂Se₃ for Hierarchical Information Processing”

“Energy efficient and robust reservoir computing system using ultrathin (3.5 nm) ferroelectric tunneling junctions for temporal data learning”

“Reservoir computing on a silicon platform with a ferroelectric field-effect transistor”

“A Compact Fully Ferroelectric-FETs Reservoir Computing Network With Sub-100 ns Operating Speed”

I believe for a Nature Communications paper, substantial benefits/differences/novelty vs. each of these papers must be sufficiently explained by the authors in their response. One benefit is the nonvolatile ferroelectric readout component, though sadly this is only implemented experimentally for the most simple task and in other cases it is only simulated due to ‘wiring complexity’. Also this scheme bears resemblance to this recent work where different memristors are used for nonlinear processing and weight storage functionality:

Zhong, Yanan, et al. "A memristor-based analogue reservoir computing system for real-time and power-efficient signal processing." *Nature Electronics* 5.10 (2022): 672-681.

The formatting of this work unfortunately makes it quite difficult to read. Text, figures and figure captions are all well-separated so any attempt to read the paper involves lots of flipping between pages. It is a very long and dense paper, and understanding it is sadly made much harder by the authors choice to separate content in this way. Please consider using normal journal style formatting in the future! It makes the reader’s life much more pleasant. 18 pages without figures seems very long for Nature Communications too?

Ultimately, there are too many unknowns to give a clear recommendation at this stage without hearing back from the authors. I am inclined towards recommending transfer to Communications Physics/Communications Engineering due to the list of similar prior studies in both memristive computing and ferroelectric computing, but the strong performance of the Henon map task means I would like to give the authors a chance to explain themselves on some key points and strengthen their argument before deciding.

Specific technical comments follow below

The authors show significant harmonic generation at relatively low frequencies. While harmonic generation can be highly useful for neuromorphic computing, the observed increase in harmonic amplitude at relatively low frequencies (30-80 Hz) and related loss of amplitude of the fundamental signal frequency suggest that there is a relatively low frequency ceiling on this scheme. This is presumably occurring as the drive frequency is approaching the physical device recovery speed.

This is much slower than with the typical memristors which are used for reservoir computing, and it looks like the all-ferroelectric scheme may not be able to operate at frequencies much in excess of 100-200 Hz? I think many readers will have similar questions & the authors should comment on the upper frequency bound imposed by this harmonic distortion.

Fig 3f) feels redundant, the Fourier transform of an input sine-wave can not be anything other than a delta like function at the fundamental frequency?

‘or the currents acquired through additional read operations can be used’ – it is unclear from the text how this differs from the previous statement of ‘directly measure current responses’. Please rephrase such that the general reader can understand the difference between these choices.

For the curvature discrimination task, the text states ‘These reservoir states, after being converted to voltage signals, are applied the readout network whose weights are trained offline’. How are these weights trained, via linear regression across the memristor currents in response to a training set? This is never stated and it’s a crucial detail. If so, how long was the training set? How long was the test set etc? You state that $N \times M$ weights are calculated and stored on the nonvolatile FD, here M is one (does this mean you just have a single nonvolatile FD?) and N is 3, so how are 3 weights stored simultaneously on a single nonvolatile FD assuming the weight for each volatile FD is different?

Key question on how much your computing results arise from your preprocessing, linear regression and nonlinear postprocessing/activation functions:

I have an ongoing issue where neuromorphic computing results are presented with no attempt at a control experiment. From the results/figures in your paper, it is hard to know how much is your device actually accomplishing, and how much of the classification accuracy comes from your curvature pre-processing (chopping it into 3 fixed amplitude pulses) and the different offline-trained weights and the sigmoidal activation function. It looks to me that if you removed the FD devices from the setup, and replaced them with linear resistors you would very likely still get strong classification. Your curvatures are pre-processed twofold: an initial chop into 3 ‘beginning, middle and end’ regions and then again into $t_{1,2,3}$ subregions. As each of your beginning, middle and end regions are sent to the same volatile FD each time, your offline weight training can apply separate weights to the different key time regions. As such, the history dependence which would ideally arise from your physical system is in reality being handled at least in part by your pre-processing. I have a feeling that replacing your volatile FDs with 3 linear resistors, and using the same preprocessing and a simple linear regression scheme, you would be able to find a combination of weights such that a curve dipping in the middle reliably results in a lower net output vs a curve rising in the middle. As long as there is some small separation between these two curve shapes, an aggressive sigmoid fitted to the small current separation between the different curve-shape output cases should be enough to force your ‘neuronal output’ to 0 or 1.0. I have a strong feeling that for such a simple task, the FDs are probably not doing as much as it initially appears.

It may of course be that your volatile FDs accomplish this in a more impressive way, or I may indeed be missing some subtlety and linear resistors would not quite manage this task - but I believe that this subtlety of how powerful your multi-step preprocessing, linear regression and then artificial nonlinear post-processing are will likely go unnoticed by readers who are not themselves active in the field of neuromorphic computing and may indeed provide a false or exaggerated sense of what your FD devices are actually accomplishing. An example of how much is actually being accomplished by pre/post-processing can be found in “Abreu Araujo, Flavio, et al. "Role of non-linear data processing on speech recognition task in the framework of reservoir computing." *Scientific reports* 10.1 (2020): 1-11.”

Have the authors tried a control experiment here? How do the results look when bypassing the physical system? I would like evidence of a control experiment demonstrating what value the FDs bring to the computing scheme. Additionally no references are given to relate this task to anything in

the research literature, so again it is very challenging to assess the computational merit of your system here in relation to other schemes or pure software.

The digit recognition task is somewhat more impressive. However - again, there is the issue of how far could simple weight regression go without the FDs? The 1 = input something, 0 = input nothing scheme would still allow substantially different traces for fig 5c once iterated across the five rows of the digit, and again I would not be surprised if substantial recognition was achieved.

Given that the devices are all electrical and presumably easy to run large datasets through, one wonders why the 'industry standard' MNIST 28x28 pixel handwritten digit task was not implemented instead? Presumably it is because the FDs struggle to differentiate sequences of beyond 3 pulses? I am sure the authors attempted more complex tasks, the fact that this very limited version of digit recognition is presented does suggest that the physical neuromorphic computing scheme shown here is somewhat limited vs. the state of the art. 10 digits (0-9) are shown to the FD system for training, then a further 12 digits are evaluated for testing. The test set is somewhat short for such a high impact journal, it would be good to see how the system fares at recognising a larger set of other digits.

The authors could offer some comment on why this non-standard digit recognition task was chosen, and I think if they are attempting publication in a journal such as Nature Communications it is fair to ask to see control experiments for the curvature discrimination and longer test/train sets for image recognition.

Crucially, there is the following confusing statement:

"However, due to the relatively large size of the readout network (6×10 ; including biases), it was difficult to implement it with the nonvolatile FDs using the wiring method (see Methods). Software-based implementation was therefore employed, where the experimentally measured conductance values in the LTP/LTD curves (Fig. 2d) were used for the simulation of weights. Hereafter unless otherwise specified, the readout network was always simulated based on the experimental data"

The authors must clarify this, do they mean that the I/V curves of the volatile FDs are just being simulated as a lookup table, replacing a typical software ReLU / Sigmoid activation function with the experimentally recorded I/V characteristics and performing all learning in simulation? Or do the authors mean that they actually ran different current pulses through their volatile FDs for each digit, and just performed offline learning instead of using real nonvolatile FDs? This is a key statement as much of the novelty and merit of this scheme is in the combination of volatile and nonvolatile FDs. Removing the nonvolatile FDs already limits impact of the results somewhat.

The Henon map performance is quite impressive, again a control would be nice showing the effect of linear regression on a 48 virtual-node system with linear elements instead of FDs, or varying the number of virtual nodes/parallel channels to give a clearer impression of how much performance is coming from the offline virtual multiplexing and how much from the specific choice of ferroelectric memristor. However, the performance is definitely impressive – the authors have done a good job here. This raises the question, how is the Henon map performance so strong when the curvature and digit recognition is not particularly impressive? It may well be that the harmonic generation/temporal nonlinearity of the reservoir is excellently suited to such chaotic prediction

tasks, in which case this is a nice result of the paper – but it could be better explained relative to the other results and other systems in the literature.

It is nice that the authors give a number for energy consumption, however they make a fair number of assumptions and simplifications here (i.e. disregarding the rest of their circuitry) so its hard to know how much to trust it. It does sound impressive though.

Reviewer #2 (Remarks to the Author):

In this paper the authors present a new way to exploit ferroelectric devices, by taking advantage of their RC characteristics. For this purpose they use the principle of reservoir computing. The principle of reservoir computing is that the network is connected in a way that the temporal aspect of the inputs and the capacity to memorize them over time is crucial. What the authors do is to apply this principle to the devices they have in a crossbar structure and they demonstrate experimentally the interest of their approach. The demonstration is done on three distinct tasks: in a first time a first task on curvature, then a recognition of figure, a classification of waveform and finally a henon map prediction. The latter is the most interesting indeed the classification required a complex dynamic system that fully exploits the complexity of the system.

It is a well written article, with a clear message, precise information, clear and well-presented figures. The overall article is very clear. The general idea of the paper is quite easy to understand and innovative. The experimental demonstration is quite impressive with a complete set-up, a perfectly working system. This set-up allows a number of experiments to be carried out, which has made it possible to perform different classification tasks. The classification tasks have been chosen in an appropriate way by exploiting the system up to its greatest interest which is that of its use through the temporality of the RC type circuit.

Nevertheless, there are still some questions to answer concerning the fundamental interest of the paper. Especially as the experiments are performed only with a few devices and the projection are made only on simulation. The first question concerns the variability, in the article the authors present several programming levels resulting in several resistance values which allows the system to have several operating time ranges for the different values of RC constants. But, there is no cycle-to-cycle or neither device-to-device variability measurements, which I think is necessary for papers that use new memory devices. It would be nice also to have a more in-depth study on these variabilities, will the variability of the devices tend to diminish a lot the performances? To what extent do the distributions overlap with each other? The second question concern the algorithm. Did the authors tried to look at other approaches such as e-prop (Bellec et al.). In this approach, the dynamic of

spikes is used for a learning process. Whereas, reservoir computing is a quite old paradigm that doesn't scale to complex task.

Reviewer #3 (Remarks to the Author):

In this manuscript, the authors used combination of Ferroelectric thin films (Volatile and nonvolatile ferroelectric diodes) to realize the reservoir computing devices. The manuscript is well written with clear storyline, and the all-ferroelectric RC system has demonstrated good working function.

I recommend publication of this manuscript in Nat. Commun, after authors clarify below questions.

1 The imprint from the P-V loop for 19 Pa film is not persuasive enough. As the imprint is related with the gap. I suggest the authors to use PFM to provide evidence of the imprint from phase or amplitude loops, as compared with the non-volatile 15 Pa films.

2. The authors explains the downward imprint electric field due to the surface accumulated oxygen vacancies according to the XPS data. However, the drifting in 19Pa samples is too small to make the conclusion. In addition, there is no explanation of why the oxygen vacancies are near surface of 19 Pa film. Are these inhomogeneous oxygen vacancies distribution stable over time or temperature or external electric field?

3. What's the domain configuration (up and down domain ratio and distributions) like for 19Pa and 15Pa film? Does the local IV match the macroscopic resistive switching (on electrode), especially at up and down domain respectively.

Responses to Reviewers' Comments

Reviewer #1

Overall evaluation: The authors present a thorough study on an all ferroelectric memristor computation system, split roughly into two parts: description of the material synthesis and device architecture and discussion of some benchmark reservoir computation tasks. The work seems solid if a little difficult to understand at times due to its long, dense nature, formatting issues and lack of clarity on some key points (see below). Overall it is a nice work, some important clarifications need to be made before it is possible to recommend it for publication in Nature Communications.

Reply: We thank the reviewer for the thorough assessment and valuable comments, which are helpful to improve the manuscript. Our responses to the specific comments are presented below.

Comment (1): I am not a ferroelectric scientist by training, and while the material system results certainly look nice, thorough and well-presented – I lack the grounding in the literature to assess whether there is strong novelty in the design or implementation of the ferroelectric memristors used here. The authors could make a better attempt at situating this section within the literature for the broad readership at the multidisciplinary journal they are submitting to.

Reply: We have revised the Introduction to show the novelties of our all-ferroelectric RC system and its building blocks — ferroelectric diodes (FDs). We believe that the revised Introduction is more appealing to the broad readership of Nature Communications.

In the third paragraph of Introduction, we first summarize the issues of the mainstream RC hardware systems based on filamentary memristors. To address these issues, we resort to the ferroelectric-based RC system and highlight its immense potential advantages. The revised third paragraph of Introduction is presented below:

“Notably, both the diffusive and drift memristors were mainly based on a filamentary mechanism, which can, however, lead to relatively large variations and low endurance due to the stochasticity of filament formation/rupture processes. This limits the prediction accuracy and reliability of the filamentary memristor-based RC system. Compared with filamentary switching, ferroelectric polarization switching is a more deterministic switching mechanism²⁸. Ferroelectric memristors, which use the polarization switching to tune the resistance^{28,29}, can thus exhibit highly reproducible memristive responses and potentially unlimited endurance³⁰⁻³⁴. Besides, they also show high switching speed and low power consumption³⁵⁻³⁷. Using ferroelectric memristors as building blocks may therefore facilitate the development of highly reliable, accurate, fast, and energy-efficient ferroelectric-based RC systems.”

In the fourth paragraph of Introduction, we point out that so far there has been no demonstration of an all-ferroelectric RC system, and attribute the reason to the limitations of the existing ferroelectric memristors used in RC systems. This necessitates the exploration of more suitable ferroelectric memristors for all-ferroelectric RC systems. The revised fourth paragraph of Introduction is presented as follows:

“However, the use of ferroelectric memristors in RC systems is currently scarce and mainly restricted to the reservoir^{11,27,38-42}, as summarized in Supplementary Table S1. All-ferroelectric implementation of a whole RC system still remains undemonstrated. The reason for this is probably because the ferroelectric memristors used hitherto in RC systems — ferroelectric tunnel junction (FTJ)²⁷ and ferroelectric field-effect transistor (FeFET)^{11,40,42} — possess inherently large depolarization fields (E_{dp}) arising from ultra-small ferroelectric film thickness^{43,44} and poor screening at ferroelectric/semiconductor interface⁴⁵, respectively. This makes them voluntary to exhibit volatile characteristics while difficult to be engineered into nonvolatile memristors to implement the readout network.”

In the fifth paragraph of Introduction, we highlight the novelty of using FDs as the building blocks of the all-ferroelectric RC system. The revised fifth paragraph of Introduction is presented as follows:

“To construct an all-ferroelectric RC system, alternative ferroelectric memristors capable of being engineered into both volatile and nonvolatile memristors (for the reservoir and readout network, respectively) are demanded. A promising candidate is the ferroelectric diode (FD) which operates by using polarization to modulate the interfacial Schottky barrier⁴⁶⁻⁴⁸. FD is inherently subjected to a much smaller E_{dp} compared with FTJ and FeFET, because it comprises a relative thick ferroelectric film (several tens to hundreds of nanometers) sandwiched between two metal electrodes with good screening ability. Consequently, FD can readily function as a nonvolatile memristor^{29,47,48}. In addition, FD can also be engineered to be volatile by judiciously introducing certain mechanisms for polarization back-switching⁴⁹⁻⁵¹. Therefore, it is quite promising to use appropriately

engineered FDs to implement both the reservoir and readout network, thus realizing an all-ferroelectric RC system in hardware (Fig. 1b-d).”

Comment (2): My expertise here is on the neuromorphic computing side, and it must be said that the Henon map results are very impressive, the authors have done a commendable job at this challenging task. However, it is not explained why the Henon map performance is so good, especially when the other computing results are not particularly impressive and are far from cutting edge. Is it because of some new physics on show here? as the reservoir architecture is fairly standard. Unfortunately this is not really commented on, so it’s hard to know whether it just so happens that the dynamics of the system here are a good match for the Henon attractor (this is a known phenomena in neuromorphic computing, each chosen physical system will be better or worse suited to given nonlinear problems) or if there is some exceptional computing going on.

In trying to ascertain this, the reader’s job is complicated by a lack of control experiments to indicate how much of the observed computing performance is achieved via pre-processing, virtual nodes and offline nonlinear post-processing, and how much is achieved via the unique qualities of the ferroelectric system. For me, gaining understanding of this is the deciding factor on whether this work is fit for Nature Communications or if it would be a better fit for Communications Physics/Communications Engineering.

Reply: We have performed control experiments and further analyses following your suggestion, as detailed as follows.

(a) Role post-processing

In the experiment of Hénon map prediction, the readout network was trained with linear regression. No nonlinear post-processing was performed. There is thus little contribution from the post-processing to the RC performance, which can be evidenced by a control experiment shown in Section (b).

(b) Role of pre-processing (linear signal conversion)

The pre-processing mainly involves two processes: 1) the mask process which can generate virtual nodes, and 2) the linear conversion of input signals to pulse voltages. The role of the mask process will be discussed later. The linear conversion of input signals to pulse voltages is a widely used way to pre-process the time-series data [1-3], and such linear signal conversion may contribute little to the RC performance [please refer to Sci. Rep. 10, 328 (2020), i.e., the paper recommended by the reviewer in Comment (9)].

Figure R1. (a) Schematic of a control RC system where the reservoir and mask process are removed. (b) Predicted time series versus ideal targets obtained from the RC system shown in a.

To confirm the minor roles played by the pre-processing of linear signal conversion and the post-processing of linear regression, we have performed a control experiment. The input signals $x(n)$ and $x(n - 1)$, after the linear signal conversion, were directly fed to a

readout network which was trained by the linear regression [see Figure R1(a)]. Neither mask process nor reservoir was used. The predicted time series are shown in Figure R1(b), which deviate significantly from their corresponding ideal targets. The NRMSE value on the test set is further calculated to be 0.98, which is extremely high. These results demonstrate that the pre-processing of linear signal conversion and the post-processing of linear regression are minor factors contributing to the RC performance.

(c) Role of mask process

The mask process is known to be capable of improving the RC performance since it can generate virtual nodes to effectively expand the reservoir size [1,3]. To demonstrate this in our RC system, the RC performance was investigated with varying number of masks (n_m) and mask length (l_m). In the experiment of Hénon map prediction, each mask sequence is processed by one volatile FD, generating l_m virtual nodes. The reservoir size, or the total number of virtual nodes, is thus $l_m \times n_m$.

Figure R2. NRMSE on the test set as a function of (a) number of masks n_m (mask length l_m fixed at 3), and (b) mask length l_m ($l_m \times n_m$ fixed at 24).

Figure R2(a) shows that NRMSE decreases with increasing n_m (l_m fixed at 3). This is within expectation because the reservoir size becomes larger when using more masks. More reservoir states are thus generated, which can help to better capture the features of the input signals. In addition, as more masks are used, more volatile FDs are also used correspondingly. The device-to-device variation of the volatile FDs can help to expand the effective reservoir size, which is an additional factor contributing to the decrease of NRMSE.

Then, the effect of mask length l_m was investigated by varying l_m while fixing $l_m \times n_m$ at 24. Figure R2(b) shows that NRMSE first decreases and then increases with increasing l_m , and it reaches the minimum value of 0.017 at $l_m = 3$. When the mask length is too short, e.g., $l_m = 1$, the number of different types of mask sequences is very small. Hence, the richness of reservoir states is very low, causing a high NRMSE. Increasing l_m can improve the reservoir state richness, thus reducing NRMSE. However, as l_m further increases, the decrease of feedback strength becomes dominant [3]. This is the main cause for the rise of NRMSE with increasing l_m . In addition, the role of the device-to-device variation becomes weaker as l_m increases (given that $l_m \times n_m$ is fixed at 24, a larger l_m leads to a smaller n_m). This is an additional factor contributing to the rise of NRMSE.

As Figure R2(b) shows that $l_m = 3$ is the optimal mask length under the constraint of $l_m \times n_m = 24$, we therefore report the results obtained at $l_m = 3$ and $n_m = 8$ in the main text. However, it should be noted that the optimal value of l_m is task-dependent, and it may change if different pulse parameters and different constraints of $l_m \times n_m$ are used.

The above results have demonstrated that the mask process contributes to the RC performance; however, it indeed plays an auxiliary role. The decisive factors are actually the device characteristics of the volatile and nonvolatile FDs, as demonstrated as follows.

(d) Roles of device characteristics of volatile and nonvolatile FDs

For an RC system, the nonlinearity and short-term memory of the reservoir and the multilevel nonvolatile weights of the readout network are of critical importance to the performance. These functional requirements of the reservoir and readout network are well fulfilled by our volatile and nonvolatile FDs, respectively.

For the nonvolatile FD, it is engineered to be free from imprint field (E_{imp}), so it can exhibit good polarization stability and consequent nonvolatile memristive switching. The multilevel nonvolatile conductance states (>4 bits) can be used to precisely map the weights in the readout network, which is apparently critical for the RC performance.

On the other hand, the volatile FD with purposely introduced E_{imp} is used to implement the reservoir. Thanks to the complex polarization dynamics (including the nonlinear, history-dependent polarization switching under external field and spontaneous polarization back-switching induced by E_{imp}), the volatile FD well exhibits the nonlinearity and short-term memory as required by a reservoir. To demonstrate that the volatile FD-based reservoir is essential to the RC performance, we replaced the volatile FDs in the reservoir with linear resistors (without changing other factors like the mask process and the readout network) [see Figure R3(a)] and investigated how the performance would change. As shown in Figure R3(b), the RC system using a linear resistor-based reservoir yields rather poor predicted results in the Hénon map prediction task. Its NRMSE value on the

test set is as high as 0.70, which is much higher than the value of 0.017 achieved by the counterpart using a volatile FD-based reservoir. This unambiguously demonstrates that the volatile FD-based reservoir is the key to the RC performance.

Figure R3. (a) Schematic of a control RC system where the volatile FDs in the reservoir are replaced by linear resistors. (b) Predicted time series versus ideal targets obtained from the RC system shown in a.

Note that in the above RC system using a linear resistor-based reservoir, although the mask process ($l_m = 3$ and $n_m = 8$) is used, the performance is still poor. This is because the linear resistors have no memory effect, causing the virtual nodes generated by the mask process to be independent of each other. Hence, the linear resistor-based reservoir is unable to capture the features in the temporal inputs, even though the mask process is used. This in turn suggests that the mask process alone could not result in good RC performance.

In fact, the mask process plays an auxiliary role, i.e., expanding the reservoir size. It works only under the condition that the devices in the reservoir (like our volatile FDs) possess nonlinearity and short-term memory. With such device characteristics, the virtual nodes are nonlinearly coupled, and the current state of a virtual node depends on its own

previous state, the current state of its neighboring nodes, and the input signal applied to this node. The reservoir is hence capable of capturing both local temporal features within an input window and more global features among input windows, leading to good RC performance.

(e) Physical mechanisms underlying the device characteristics

As demonstrated above, the good performance of our all-ferroelectric RC system in the Hénon map prediction can be attributed to the following factors: the device characteristics of the volatile and nonvolatile FDs (essential) and the mask process (auxiliary). Below we will further analyze the physical mechanisms underlying the device characteristics, so as to gain deeper insights into why the performance is so good.

As shown in Eqs. (1) and (2) in the main text, the Hénon map prediction is a nonlinear 2D mapping problem, where each output depends on the recent past results but not on the far past. Such problem is well suited for reservoirs based on memristors with nonlinearity and short-term memory, like our volatile FDs.

The nonlinearity of our volatile FD mainly originates from both the nonlinear polarization switching and the nonlinear polarization-controlled conduction behavior. The polarization switching typically involves two microscopic processes: domain nucleation and domain growth, both of which have strong nonlinear dependencies on the applied voltage [4]. Additionally, the domains in the volatile FD are observed to be tiny and irregularly-shaped [see Figure R11 in the response to Reviewer #3's Comment (1)], which typically results in a wide distribution of switching voltages. This could allow the nonlinear polarization switching to occur in a relatively wide range of voltages, covering the voltages

applied in the Hénon map prediction task (the applied pulse voltages are within -2 V, and -2 V is around the coercive voltage where strong nonlinearity exists). Besides the nonlinear polarization switching, the nonlinear polarization-controlled conduction behavior, where the current is nonlinearly dependent on the polarization-controlled Schottky barrier height and the applied voltage [5], further adds to the device's nonlinearity.

In terms of the short-term memory in our volatile FD, it mainly originates from both the history dependence of polarization switching and the spontaneous polarization back-switching induced by E_{imp} . Some previously reported ferroelectric devices used only the history dependence of polarization switching to realize the short-term memory [4]; however, the memory effect may degrade or even disappear when the polarization is approaching saturation. This issue does not exist in our volatile FD because it also exhibits the spontaneous polarization back-switching induced by E_{imp} besides the history dependence of polarization switching. These complex polarization dynamics lead to a short-term memory effect with time constants in the millisecond scale. Accordingly, the pulse intervals used in the Hénon map prediction task are designed to match well with the time constants, ensuring the effectiveness of the short-term memory.

On the other hand, the nonvolatile FD-based readout network is also an important part of our RC system. The nonvolatile FD exhibits good polarization stability because of the absence of E_{imp} . Owing to this and the polarization-controlled conduction behavior, the nonvolatile FD exhibits multilevel nonvolatile conductance states (>4 bits), which can be used to precisely map the weights in the readout network. This is the major contribution from the nonvolatile FD to the RC performance.

In short, the device characteristics contributing to the good performance in the Hénon map prediction and their physical mechanisms can be summarized as follows. 1) For the volatile FD for the reservoir, the nonlinearity and short-term memory are the most important device characteristics. The nonlinearity mainly originates from both the nonlinear polarization switching and the nonlinear polarization-controlled conduction behavior. In particular, the tiny and irregularly-shaped domains allow the nonlinear polarization switching to occur in a relatively wide range of voltages, covering the voltages applied in the Hénon map prediction task. On the other hand, the short-term memory mainly originates from both the history dependence of polarization switching and the spontaneous polarization back-switching induced by E_{imp} . The pulse intervals used in the Hénon map task prediction are designed to match well with the time constants of memory decay, ensuring the effectiveness of the short-term memory. 2) For the nonvolatile FD for the readout network, a sufficiently large number of conductance states with good retention are the most important device characteristics. Owing to the nonvolatility of polarization in absence of E_{imp} and the polarization-controlled conduction behavior, our nonvolatile FD exhibits multilevel nonvolatile conductance states (>4 bits), which can be used to precisely map the weights in the readout network.

The above results and discussion have been added into the revised SI. Please refer to Figures S26 and S27, and Supplementary Note 2 in the revised SI.

Comment (3): The concept is not novel – memristive reservoir computing has been around for some time, as has ferroelectric reservoir computing. I would like more concrete comparisons to other ferroelectric computing systems specifically to ascertain where the

novelty lies and how novel and important these results are. Particularly these recent works:

“Multilayer Reservoir Computing Based on Ferroelectric α -In₂Se₃ for Hierarchical Information Processing”

“Energy efficient and robust reservoir computing system using ultrathin (3.5 nm) ferroelectric tunneling junctions for temporal data learning”

“Reservoir computing on a silicon platform with a ferroelectric field-effect transistor”

“A Compact Fully Ferroelectric-FETs Reservoir Computing Network With Sub-100 ns Operating Speed”

I believe for a Nature Communications paper, substantial benefits/differences/novelty vs. each of these papers must be sufficiently explained by the authors in their response. One benefit is the nonvolatile ferroelectric readout component, though sadly this is only implemented experimentally for the most simple task and in other cases it is only simulated due to ‘wiring complexity’. Also this scheme bears resemblance to this recent work where different memristors are used for nonlinear processing and weight storage functionality: Zhong, Yanan, et al. "A memristor-based analogue reservoir computing system for real-time and power-efficient signal processing." *Nature Electronics* 5.10 (2022): 672-681.

Reply: We have made a thorough comparison between our work and other recent works to highlight the novelties of our work.

(a) Novelty 1: realizing an all-ferroelectric RC system for the first time

Table R1. Comparison of the key features of the ferroelectric-based RC systems reported in recent works and ours.

Reservoir system	Reservoir			Readout network	Circuit-level demonstration?	Power consumption (for device in reservoir only)	Accuracy (MNIST)
	Devices	Polarization dynamics clearly revealed?	Richness of polarization dynamics				
Ref. [6]	Hf _{0.5} Zr _{0.5} O ₂ -based FTJ	No	Medium	RRAM	Yes	~70 μ W	92.3%
Ref. [4]	Hf _{0.5} Zr _{0.5} O ₂ -based FeFET	No	Low	Simulation	No	~900 μ W	–
Ref. [7]	Hf _{0.5} Zr _{0.5} O ₂ -based FeFET (volatile)	No	Medium	Hf _{0.5} Zr _{0.5} O ₂ -based FeFET (volatile)	No	~3000 μ W	95.1%
Ref. [8,9]	α -In ₂ Se ₃ -based FeFET	No	Medium	Simulation	No	~60 μ W	86.1%
Ours	BiFeO ₃ -based FD (volatile)	Yes	High	BiFeO ₃ -based FD (nonvolatile)	Yes	~11.8 μ W	89.5%

Table R1 summarizes the key features of the ferroelectric-based RC systems reported in recent works. Two facts can be extracted:

(i) The ferroelectric-based RC systems in most of the recent works used ferroelectric tunnel junctions (FTJs) and ferroelectric field-effect transistors (FeFETs) to implement only the reservoirs, while the readout networks were either simulated or implemented with a mature RRAM chip. Only the work in Ref. [7] attempted to use a Hf_{0.5}Zr_{0.5}O₂ (HZO)-based FeFET to implement the readout network, but this FeFET, identical that used for the reservoir, was indeed volatile and failed to meet the functional requirements of the readout network.

(ii) The ferroelectric-based RC systems in most of the recent works were demonstrated at the device level. Only the work in Ref. [6] reported a circuit-level demonstration, but the readout network was implemented with RRAM rather than ferroelectric devices.

From the above two facts, it is noted that so far there has been no demonstration of an all-ferroelectric RC system (particularly at the circuit level). This is disappointing because the all-ferroelectric RC system promises higher performance and robustness than the existing RRAM-based RC systems [see reasons in the response to Comment (1)]. In this work, we develop an all-ferroelectric RC system, where the reservoir and readout network are implemented with the volatile and nonvolatile FDs, respectively, and demonstrate its temporal signal processing capability at the circuit level (albeit with simple tasks). Therefore, realizing an all-ferroelectric RC system for the first time is the first novelty of this work.

In response to the reviewer's comment on the nonvolatile FD-based readout network which is experimentally implemented only for the simplest task (i.e., the curvature discrimination), we admit that this experiment is very preliminary and it is used only to demonstrate the feasibility of the all-ferroelectric RC system. Scaling up the all-ferroelectric RC system for complex tasks is currently unavailable, because the fabrication of large-scale arrays is still challenging for emerging memory devices like our FDs. Nevertheless, we believe that with the maturation of fabrication processes, large-scale all-ferroelectric RC systems for complex tasks can eventually be realized.

(b) Novelty 2: the use of FDs as building blocks

The second novelty of this work is the use of FDs as the building blocks of an RC system. Previously used FTJ and FeFET possess inherently large depolarization fields (E_{dp}) because of ultra-small ferroelectric film thickness and poor screening at ferroelectric/semiconductor interface, respectively. This makes FTJ and FeFET voluntary to exhibit volatile characteristics while difficult to be engineered into nonvolatile memristors to implement the readout network. By contrast, FD is inherently subjected to a much smaller E_{dp} compared with FTJ and FeFET, because it comprises a relative thick ferroelectric film (several tens to hundreds of nanometers) sandwiched between two metal electrodes with good screening ability. Therefore, FD can readily function as a nonvolatile memristor. In addition, by judiciously introducing an E_{imp} without changing the device structure, the FD is engineered to be volatile. The volatile and nonvolatile FDs are further used to implement the reservoir and readout network, respectively, eventually forming a well-functioning RC system.

Using FDs as the building blocks is therefore the key to the successful implementation of the all-ferroelectric RC system, which would provide great inspiration for researchers working on ferroelectric-based neuromorphic computing. Additionally, it is noteworthy that the derivation of the volatile and nonvolatile FDs from the same device structure has not been realized in other types of ferroelectric memristors.

(c) Novelty 3: clearly revealed polarization dynamics and high richness of polarization dynamics

The third novelty of this work is that the polarization dynamics of our volatile FD are clearly revealed, and their richness is in principle higher than those of other ferroelectric memristors used for reservoirs.

As shown in Table R1, the polarization dynamics remain largely unclear in previously used FTJs and FeFETs for reservoirs. This is because it is difficult or even impractical to directly measure polarization in these devices owing to either large leakage current or poor charge screening. By contrast, FD allows the direct polarization measurement because of the suppressed leakage current (arising from the relatively thick ferroelectric film and the reverse-bias Schottky barrier) and the good charge screening provided by the metal electrodes [10]. Polarization dynamics of our volatile FD have thus been systematically studied [Figures 2(b) and 3(b) in the main text and Figures S4, S5(a)-(c), S9, and S14(a)-(b) in SI]. In particular, the E_{imp} -induced polarization back-switching is clearly revealed in our volatile FD [Figure 3(b) in the main text and Figure S9 in SI], which could greatly enrich the polarization dynamics.

Table R1 also presents that the richness of polarization dynamics of our volatile FD is in principle the highest, as explained as follows. For the $\text{Hf}_{0.5}\text{Zr}_{0.5}\text{O}_2$ (HZO)-based FeFET [4], it exhibits only the nonlinear, history-dependent polarization switching behavior, which is common for all the ferroelectric devices. It therefore has the lowest richness of polarization dynamics. For other FTJs [6] and FeFETs [7-9], although the E_{dp} -induced polarization decay was claimed to exist in these devices, it was not unambiguously revealed because of the difficulty in directly measuring the polarization, as mentioned earlier. Even if the E_{dp} existed, it was only able to reduce the polarization toward zero, while unable to reverse the direction of polarization. For our volatile FD, the presence of E_{imp} and the E_{imp} -induced polarization back-switching are clearly revealed. Moreover, E_{imp} could even reverse the direction of polarization [Figure S9(c) in SI]. Therefore, the richness

of polarization dynamics of our volatile FD is in principle higher than those of previous devices with E_{dp} .

(d) Novelty 4: low power consumption

The fourth novelty of this work is the low power consumption of our all-ferroelectric RC system. Table R1 compares the power consumptions of different ferroelectric memristors used for reservoirs. Our volatile FD exhibits the lowest power consumption of $\sim 11.8 \mu\text{W}$, well due to the relatively low operation voltage and suppressed leakage current, as mentioned earlier. Such power consumption is even at least 3 times lower than those of the state-of-the-art filamentary memristors for reservoirs [1,3,11]. Also note that the power consumption of our nonvolatile FD for readout network is even lower, reaching $\sim 126 \text{ nW}$.

(d) Novelty 5: high prediction performance

In terms of the prediction performance, although the counterparts in Ref. [6] and [7] achieve higher accuracies than our all-ferroelectric RC system in the MNIST handwritten digit recognition task (see Table R1), they use either multilayer readout networks [6] or more complex pre-processing [7], making the direct comparison between these accuracies unfair. In fact, the 89.5% accuracy achieved by our all-ferroelectric RC system is 6.5% higher than that achieved by a pioneering diffusive memristor-based RC system [12], whose image pre-processing approach, reservoir architecture, and readout network size are similar to ours.

In addition, our all-ferroelectric RC system achieves an ultralow NRMSE value of 0.017 in the Hénon map time-series prediction (see Figure 6 in the main text), which also underscores its high prediction performance.

Then, we compare our work with that by Zhong et al. [13]. Their work reported a fully analogue RC system that used dynamic memristors for the reservoir layer and nonvolatile memristors for the readout layer. The key innovation of their work was the fully analogue architecture where no digital units (such as ADCs) were needed to convert the analogue reservoir states into digital signals for the readout layer. In terms of the devices, both the dynamic and nonvolatile memristors were based on the RRAM devices.

However, the RRAM devices are known to suffer from relatively large variations and low endurance, due to the stochasticity of filament formation/rupture processes. By contrast, we use FDs with a more deterministic switching mechanism (i.e., polarization switching) as the building blocks, and construct an all-ferroelectric RC system with high prediction accuracy, high reliability, and low power consumption. Therefore, our work has made a device innovation for the hardware implementation of RC, which is different from the architecture innovation of the Zhong et al.'s work.

The above results and discussion have been added into the revised SI. Please refer to Table S1 and related discussion in the revised SI.

Comment (4): The formatting of this work unfortunately makes it quite difficult to read. Text, figures and figure captions are all well-separated so any attempt to read the paper involves lots of flipping between pages. It is a very long and dense paper, and understanding it is sadly made much harder by the authors choice to separate content in this way. Please consider using normal journal style formatting in the future! It makes the reader's life much more pleasant. 18 pages without figures seems very long for Nature Communications too?

Reply: We have reformatted our manuscript, and prepared both the single- and double-column versions. In both versions, the figures and figure captions are placed to be close to their related texts.

In terms of our manuscript's length, the double-column version has 15 pages in total (including the figures). Similar lengths have been seen in some previously published NC papers (e.g., Ref. [14]).

Comment (5): Ultimately, there are too many unknowns to give a clear recommendation at this stage without hearing back from the authors. I am inclined towards recommending transfer to Communications Physics/Communications Engineering due to the list of similar prior studies in both memristive computing and ferroelectric computing, but the strong performance of the Henon map task means I would like to give the authors a chance to explain themselves on some key points and strengthen their argument before deciding.

Reply: We have modified the Introduction and made a thorough comparison between our work and those recommended by the reviewer, based on which the novelties of our work have been clearly revealed [please refer to our responses to Comment (1) and (3)]. We have also performed control experiments and confirmed that the device characteristics of our volatile and nonvolatile FDs are the decisive factors leading to the good Hénon map prediction performance [please refer to our response to Comment (2)].

We therefore believe that the reviewer's concerns can be well addressed with our above responses.

Comment (6): The authors show significant harmonic generation at relatively low frequencies. While harmonic generation can be highly useful for neuromorphic computing, the observed increase in harmonic amplitude at relatively low frequencies (30-80 Hz) and related loss of amplitude of the fundamental signal frequency suggest that there is a relatively low frequency ceiling on this scheme. This is presumably occurring as the drive frequency is approaching the physical device recovery speed.

This is much slower than with the typical memristors which are used for reservoir computing, and it looks like the all-ferroelectric scheme may not be able to operate at frequencies much in excess of 100-200 Hz? I think many readers will have similar questions & the authors should comment on the upper frequency bound imposed by this harmonic distortion.

Fig 3f feels redundant, the Fourier transform of an input sine-wave can not be anything other than a delta like function at the fundamental frequency?

Reply: We have tested the upper frequency bound of our volatile FD in the harmonic generation. As shown in Figure R4, higher harmonics can be generated at high input frequencies of 200, 500, and 1200 Hz. Further increasing the input frequency is currently unavailable because of the limited bandwidth of our measurement system. We can therefore only confirm that the upper frequency bound of our volatile FD is at least 1200 Hz, demonstrating its potential of high-speed operation.

Figure R4. FFT spectra of the output currents from the volatile FD which is subjected to input sinusoidal voltage waves with different frequencies.

Figure R4 has been added into the revised main text as Figure 3(f), and the original data of the output currents have been added into the revised SI (see Figure S19). Additionally, following the reviewer’s suggestion, previous Figure 3(f) which shows the FFT spectra of the input sine-waves has been removed.

Comment (7): ‘or the currents acquired through additional read operations can be used’ – it is unclear from the text how this differs from the previous statement of ‘directly measure current responses’. Please rephrase such that the general reader can understand the difference between these choices.

Reply: Sorry for the confusion. The related sentences have been modified:

“To obtain the reservoir state, there are typically two approaches: 1) measuring the current responses of the volatile FDs to input write pulses and directly using them as the reservoir state⁷, and 2) applying read pulses after input write pulses and using the read currents as the reservoir state^{6,14}. The latter approach is used in this experiment.”

Comment (8): For the curvature discrimination task, the text states ‘These reservoir states, after being converted to voltage signals, are applied the readout network whose weights are trained offline’. How are these weights trained, via linear regression across the memristor currents in response to a training set? This is never stated and it’s a crucial detail. If so, how long was the training set? How long was the test set etc? You state that $N \times M$ weights are calculated and stored on the nonvolatile FD, here M is one (does this mean you just have a single nonvolatile FD?) and N is 3, so how are 3 weights stored simultaneously on a single nonvolatile FD assuming the weight for each volatile FD is different?

Reply: Please see the experimental details below.

(a) Method to train the readout weights

The readout weights were trained via the logistic regression using the volatile FDs’ responses to the training set as the inputs.

(b) Samples in the training and test sets

Figure R5. (a) 10 curves included in both the training and test sets. (b) Another 2 curves included in the test set only.

In our previous manuscript, the training and test sets contain the same samples, i.e., the 10 curves shown in Figure R5(a). Half of these curves have positive curvature while the rest have negative curvature.

In the revised manuscript, we have added two new curves into the test set, as shown in Figure R5(b). These two curves are correctly classified by our all-ferroelectric RC system, which is unrealizable for a linear resistor-based RC system. Please see our response to Comment (9) for details.

(c) Number of devices in the RC system

In fact, $N \times M$ rather than M represents the number of weights. M is the number of output neurons. In the curvature discrimination experiment, 3 volatile FDs were used for the reservoir, and 4 nonvolatile FDs were used for the readout network. The 4 nonvolatile FDs are sufficient to store 3 weights ($N \times M = 3$) and 1 bias.

The above details have been added into the “All-ferroelectric RC system and its application in curvature discrimination” section in the revised main text.

Comment (9): Key question on how much your computing results arise from your preprocessing, linear regression and nonlinear postprocessing/activation functions: I have an ongoing issue where neuromorphic computing results are presented with no attempt at a control experiment. From the results/figures in your paper, it is hard to know how much is your device actually accomplishing, and how much of the classification accuracy comes from your curvature pre-processing (chopping it into 3 fixed amplitude pulses) and the different offline-trained weights and the sigmoidal activation function. It looks to me that if you removed the FD devices from the setup, and replaced them with

linear resistors you would very likely still get strong classification. Your curvatures are pre-processed twofold: an initial chop into 3 'beginning, middle and end' regions and then again into t1,2,3 subregions. As each of your beginning, middle and end regions are sent to the same volatile FD each time, your offline weight training can apply separate weights to the different key time regions. As such, the history dependence which would ideally arise from your physical system is in reality being handled at least in part by your pre-processing. I have a feeling that replacing your volatile FDs with 3 linear resistors, and using the same preprocessing and a simple linear regression scheme, you would be able to find a combination of weights such that a curve dipping in the middle reliably results in a lower net output vs a curve rising in the middle. As long as there is some small separation between these two curve shapes, an aggressive sigmoid fitted to the small current separation between the different curve-shape output cases should be enough to force your 'neuronal output' to 0 or 1.0. I have a strong feeling that for such a simple task, the FDs are probably not doing as much as it initially appears.

It may of course be that your volatile FDs accomplish this in a more impressive way, or I may indeed be missing some subtlety and linear resistors would not quite manage this task - but I believe that this subtlety of how powerful your multi-step preprocessing, linear regression and then artificial nonlinear post-processing are will likely go unnoticed by readers who are not themselves active in the field of neuromorphic computing and may indeed provide a false or exaggerated sense of what your FD devices are actually accomplishing. An example of how much is actually being accomplished by pre/post-processing can be found in "Abreu Araujo, Flavio, et al. "Role of non-linear data processing

on speech recognition task in the framework of reservoir computing." *Scientific reports* 10.1 (2020): 1-11."

Reply: In the curvature discrimination experiment, the pre-processing mainly includes 2 steps: 1) chopping each curve into 3 sections, and 2) converting each section to a 3-timeframe pulse train. The beginning, middle, and end sections of a curve are therefore represented by 3 pulse trains, respectively. The 3 pulse trains are then applied to a reservoir consisting of 3 volatile FDs, with each device processing one pulse train.

As seen above, the key function of the pre-processing is converting the spatial information in the curve into the temporal features in the streaming inputs. How the curve is pre-processed thus affects the temporal features to be extracted by the volatile FDs, which in turn influences the classification accuracy.

For example, if the curve is not chopped, one 9-timeframe pulse train is sufficient to represent this curve (assuming that the product of the number of pulse trains and that of timeframes is 9). Correspondingly, only one volatile FD is used to process this pulse train. Due to the short-term memory of the volatile FD, only the spatial information near the end of the curve can be well captured while that in the beginning and middle may be lost, resulting in a poor accuracy. On the other hand, if the curve is chopped into too many sections, e.g., 9 sections, 9 1-timeframe pulse trains are needed to represent this curve (also assuming that the product of the number of pulse trains and that of timeframes is 9). Correspondingly, 9 volatile FDs are used to process the 9 pulse trains. Because each pulse train has only 1 timeframe, it is unable to use the memory effect in the volatile FDs. The reservoir in this case is ineffective.

Therefore, to take full advantage of the volatile FD-based reservoir, we have chopped the curve into an appropriate number of sections (i.e., 3 sections), and converted each section to a pulse train with an appropriate number of timeframes (i.e., 3 timeframes). Similar way of pre-processing has been widely used for physical RC systems when handling spatial pattern recognition tasks [2,15].

Although the pre-processing has certain effects on the RC performance, we argue that it plays an auxiliary role. The volatile FD-based reservoir indeed plays the essential role because it is responsible for extracting temporal features. This can be evidenced through a control experiment following the reviewer's suggestion.

In this control experiment, the volatile FDs in the reservoir were replaced by linear resistors [Figure R6(a)], while the approaches of pre-processing, linear regression, and nonlinear post-processing (sigmoid activation function) were kept the same as those for the volatile FD-based RC system. The current responses of the linear resistors to write pulses were recorded and directly used for reservoir states, while no additional read pulses were applied.

Figure R6. (a) Schematic of a control RC system where the volatile FDs in the reservoir are replaced by linear resistors. (b) Output currents of the volatile FDs and linear resistors in the reservoir, and Device 1-3 correspond to beginning, middle, and end sections of the curve, respectively. (c) Compared classification results of the RC systems based on the volatile FDs and linear resistors.

The linear resistor-based RC system can correctly discriminate the two classes of curves shown in Figure R5(a), which is simply because these curves have been seen by the system during the training.

To increase the difficulty of the task, two previously unseen curves, as shown in Figure R5(b), were added into the test set. As shown in Figure R6(c), the linear resistor-based RC system makes wrong classification. To understand the underlying origin, the linear resistors' current responses were further inspected. As displayed in Figure R6(b), the

linear resistor corresponding to the middle section of the curve produces a lower (higher) current when inputting the curve with negative (positive) curvature. This is the main cause for the wrong classification given the readout weights trained offline (note: it is a convention to use the readout weights obtained after training for the test, and no further adjustments to the weights is allowed during the test). Why the linear resistor produces such current responses is because it has no memory effect and hence the relative height of the last point in the middle section (i.e., the amplitude of the last pulse in the corresponding pulse train) determines the magnitude of the final current response.

By contrast, the volatile FD produces different current responses [Figure R6(b)], which is well attributed to its memory effect allowing the pulse history to influence the conductance. This in turn results in a correct classification given the readout weights trained offline, as shown in Figure R6(c).

The above compared results therefore confirm that the volatile FD-based reservoir plays an essential role in our RC system. Besides, the nonvolatile FDs, which provide multilevel nonvolatile conductance states for the mapping of readout weights, are also important for the RC performance.

The above results have been added into the revised SI (please see Figure S20 and Supplementary Note 1 in the revised SI).

In addition, we have read the paper recommended by the reviewer and studied the roles of pre-/post-processing accordingly. This paper has also been cited when discussing the roles of pre-/post-processing in our revised manuscript (see Ref. [57] in the revised main text).

Comment (10): Have the authors tried a control experiment here? How do the results look when bypassing the physical system? I would like evidence of a control experiment demonstrating what value the FDs bring to the computing scheme. Additionally no references are given to relate this task to anything in the research literature, so again it is very challenging to assess the computational merit of your system here in relation to other schemes or pure software.

Reply: We have performed a control experiment, as presented in the response to Comment (9). It is seen that replacing the volatile FDs in the reservoir with linear resistors causes wrong classification of certain curves. This demonstrates that the volatile FD-based reservoir plays an essential role in our RC system.

The curvature discrimination task is self-designed. We intend to use this simple task to demonstrate that our all-ferroelectric RC system works. To better assess the computational merit of our system, other more complex tasks like the MNIST handwritten digit recognition and Hénon map prediction can be referred to.

Comment (11): The digit recognition task is somewhat more impressive. However - again, there is the issue of how far could simple weight regression go without the FDs? The 1 = input something, 0 = input nothing scheme would still allow substantially different traces for fig 5c once iterated across the five rows of the digit, and again I would not be surprised if substantial recognition was achieved.

Reply: We have performed a control experiment following the reviewer's suggestion. In this experiment, the volatile FDs in the reservoir were replaced by linear resistors [Figure R7(a)]. The current responses of the linear resistors to write pulses were recorded and

directly used for reservoir states, while no additional read pulses were applied. Since the linear resistor has no memory effect, the last pixel in a row of a digit (i.e., the amplitude of the last pulse in the corresponding pulse train) determines the final current response of the linear resistor. As a result, it is seen from Figure R7(b) that the linear resistors corresponding to different rows produce identical final current responses when the digits “0”, “4”, “8”, and “9” from the training set [see Figure 5(a) in the main text] are presented. This means that the reservoir states corresponding to the digits “0”, “4”, “8”, and “9” are the same, making these digits indistinguishable. Similarly, the digits “5” and “6” from the training set are also indistinguishable. The linear resistor-based reservoir is therefore ineffective.

Figure R7. (a) Schematic of a control RC system where the volatile FDs in the reservoir are replaced by linear resistors. (b) Experimentally measured reservoir states after stimulating the linear resistor-based reservoir with different input images in the training set.

As a comparison, it is shown in Figure 5(d) in the main text that the volatile FD-based reservoir can produce well distinguishable reservoir states for the 10 digits in the training

set. The compared results of Figure R7 and Figure 5 in the main text therefore demonstrate that the volatile FD plays an essential role in our reservoir.

Certainly, one may use a fully-connected neural network (FCNN) for this digit recognition task. The FCNN achieves 100% and 91.7% accuracies on the training and test sets, respectively, which are the same as those achieved by our RC system. However, the number of weights needing to be trained in our RC system is only 60 (6×10 ; including biases), while that increases to 160 (16×10 ; including biases) in the FCNN. This highlights the low training cost of our RC system.

The above results have results have been added into the revised SI (please see Figure S22 and related discussion in the revised SI).

Comment (12): Given that the devices are all electrical and presumably easy to run large datasets through, one wonders why the ‘industry standard’ MNIST 28x28 pixel handwritten digit task was not implemented instead? Presumably it is because the FDs struggle to differentiate sequences of beyond 3 pulses? I am sure the authors attempted more complex tasks, the fact that this very limited version of digit recognition is presented does suggest that the physical neuromorphic computing scheme shown here is somewhat limited vs. the state of the art. 10 digits (0-9) are shown to the FD system for training, then a further 12 digits are evaluated for testing. The test set is somewhat short for such a high impact journal, it would be good to see how the system fares at recognising a larger set of other digits.

Reply: Following the reviewer’s suggestion, we have implemented the MNIST handwritten digit recognition with our RC system. All the images are pre-processed, as

schematically illustrated in Figure R8(a). An original greyscale image is first converted to a binary-pixel image. Then, the unimportant periphery area of the image is removed, reducing the image size from 28×28 to 22×20 . Each row is further chopped into 5 sections, and each section is converted to a 4-timeframe pulse train. The pulse amplitude is -2.7 V (0 V) when the pixel value is 1 (0), and the pulse width is fixed at 2 ms.

Figure R8. (a) Schematic flow of the pre-processing of a digit image and the subsequent processing by the all-ferroelectric RC system. (b) Experimentally measured reservoir states corresponding to 3 example digits “3”, “5”, and “8” from the test set. (c) Confusion matrix showing the classification results obtained from the all-ferroelectric RC system versus the target labels.

After the pre-processing, each image is converted to 22×5 pulse trains, which are subsequently fed to a reservoir consisting of 22 volatile FDs. One volatile FD is responsible

for processing 5 pulse trains. After each pulse train the device's final conductance state is read out and then its conductance is reset to the initial value through a reset pulse. 110 (22×5) read current values are thus obtained, the combination of which forms the reservoir state. The reservoir state is then fed to a $(110 + 1) \times 10$ readout network for classification, where the weights are mapped onto the experimentally measured conductance values of nonvolatile FDs by simulation. The readout network is trained offline with softmax regression, and 60000 images from the MNIST dataset are used for training. After training, the recognition accuracy of the RC system is tested with 10000 images which are not included in the training set.

As an illustrative example, Figure R8(b) shows that the reservoir states corresponding to the digits “3”, “5”, and “8” are distinctly different, evidencing the effectiveness of the volatile FD-based reservoir. Figure R8(c) shows the confusion matrix produced by our all-ferroelectric RC system on the test set. Most of the digits are correctly classified, and the recognition accuracy reaches 89.5%. This accuracy is 6.5% higher than that achieved by a pioneering diffusive memristor-based RC system [12], whose image pre-processing approach, reservoir architecture, and readout network size are similar to ours. Note that several studies reported even higher accuracies. However, they typically used more complex pre-processing approaches [3,7], different reservoir architectures [15], or multilayer readout networks [6], making it unfair to directly compare their accuracies with ours.

The above results have been added into the revised SI (please see Figure S24 in the revised SI).

Comment (13): The authors could offer some comment on why this non-standard digit recognition task was chosen, and I think if they are attempting publication in a journal such as Nature Communications it is fair to ask to see control experiments for the curvature discrimination and longer test/train sets for image recognition.

Reply: We have performed the MNIST handwritten digit recognition task following the reviewer's suggestion [please see the response to Comment (12)]. In addition, we have carried out control experiments for the curvature discrimination [please see the response to Comment (9)], digit recognition [please see the response to Comment (11)], and Hénon map prediction [please see the response to Comment (2)]. These experiments help to improve the quality of our work, and therefore we believe that it can be published in Nature Communications.

Comment (14): Crucially, there is the following confusing statement: "However, due to the relatively large size of the readout network (6×10 ; including biases), it was difficult to implement it with the nonvolatile FDs using the wiring method (see Methods). Software-based implementation was therefore employed, where the experimentally measured conductance values in the LTP/LTD curves (Fig. 2d) were used for the simulation of weights. Hereafter unless otherwise specified, the readout network was always simulated based on the experimental data" The authors must clarify this, do they mean that the I/V curves of the volatile FDs are just being simulated as a lookup table, replacing a typical software ReLu / Sigmoid activation function with the experimentally recorded I/V characteristics and performing all learning in simulation? Or do the authors mean that they actually ran different current pulses through their volatile FDs for each digit,

and just performed offline learning instead of using real nonvolatile FDs? This is a key statement as much of the novelty and merit of this scheme is in the combination of volatile and nonvolatile FDs. Removing the nonvolatile FDs already limits impact of the results somewhat.

Reply: Sorry for the confusion. We actually ran different pulses through the volatile FDs for each digit, and just performed offline learning instead of using real nonvolatile FDs. Such implementation has been adopted in the tasks of digit recognition, waveform classification, and Hénon map prediction. Only in the curvature discrimination task, both the reservoir and readout network are implemented in hardware, by using the volatile and nonvolatile FDs, respectively.

For the mentioned statement, we intend to mean that the software-computed floating-point weights are not directly used for the readout network; instead, the weights are mapped onto the experimentally measured conductance values of nonvolatile FDs. This can make the simulated readout network more relevant to the device characteristics of nonvolatile FDs.

The mentioned statement has been revised to eliminate the confusion, as shown below:

“However, due to the relatively large size of the readout network (6×10 ; including biases), it was difficult to implement it with the nonvolatile FDs using the wiring method (see Methods). The readout network was therefore simulated. Nevertheless, the software-computed floating-point weights were not directly used; instead, the weights were mapped onto the experimentally measured conductance values of nonvolatile FDs (Fig. 2d).

Hereafter unless otherwise specified, the readout network was always simulated in this way, but the reservoir was still experimentally implemented with volatile FDs.”

Comment (15): The Henon map performance is quite impressive, again a control would be nice showing the effect of linear regression on a 48 virtual-node system with linear elements instead of FDs, or varying the number of virtual nodes/parallel channels to give a clearer impression of how much performance is coming from the offline virtual multiplexing and how much from the specific choice of ferroelectric memristor. However, the performance is definitely impressive – the authors have done a good job here. This raises the question, how is the Henon map performance so strong when the curvature and digit recognition is not particularly impressive? It may well be that the harmonic generation/temporal nonlinearity of the reservoir is excellently suited to such chaotic prediction tasks, in which case this is a nice result of the paper – but it could be better explained relative to the other results and other systems in the literature.

Reply: We have performed the control experiments following the reviewer’s suggestion, and the results and detailed analyses are shown in the response to Comment (2). In brief, the good performance of our all-ferroelectric RC system in the Hénon map prediction can be attributed to the following factors: the device characteristics of the volatile and nonvolatile FDs (essential) and the mask process (auxiliary). Moreover, the physical mechanisms underlying the device characteristics are analyzed in detail, which can provide deeper insights into why the performance is so good.

We have also made a thorough comparison between our RC system with others to show its merits [please see the response to Comment (3)].

Comment (16): It is nice that the authors give a number for energy consumption, however they make a fair number of assumptions and simplifications here (i.e. disregarding the rest of their circuitry) so its hard to know how much to trust it. It does sound impressive though.

Reply: We estimated the power consumption following the method reported in previous studies [1,3]. In this method, only the power consumptions of memristors were estimated while those of periphery circuits were not taken into account. Therefore, it seems fair to compare our results with theirs.

Reviewer #2

Overall evaluation: It is a well written article, with a clear message, precise information, clear and well-presented figures. The overall article is very clear. The general idea of the paper is quite easy to understand and innovative. The experimental demonstration is quite impressive with a complete set-up, a perfectly working system. This set-up allows a number of experiments to be carried out, which has made it possible to perform different classification tasks. The classification tasks have been chosen in an appropriate way by exploiting the system up to its greatest interest which is that of its use through the temporality of the RC type circuit through the temporality of the RC type circuit.

Reply: We thank the reviewer for the recognition and valuable comments, which are helpful to improve our manuscript. Our point-to-point responses are presented as follows.

Comment (1): The first question concerns the variability, in the article the authors present several programming levels resulting in several resistance values which allows the system

to have several operating time ranges for the different values of RC constants. But, there is no cycle-to-cycle or neither device-to-device variability measurements, which I think is necessary for papers that use new memory devices. It would be nice also to have a more in-depth study on these variabilities, will the variability of the devices tend to diminish a lot the performances? To what extent do the distributions overlap with each other?

Reply: Thank you for this good suggestion. We have characterized the cycle-to-cycle (C2C) and device-to-device (D2D) variations of both nonvolatile and volatile FDs, by using both DC voltage and pulse measurements. Note that in the pulse measurements, different pulse trains were used for the nonvolatile and volatile FDs: cyclic LTP/LTD pulse trains for the former and repeated “1 0 1 0” pulse trains for the latter.

(a) C2C variations of nonvolatile and volatile FDs

Figures R9(a) and (b) show the multi-cycle I - V curves of the nonvolatile and volatile FDs, respectively, as measured by the cyclic DC voltage sweeps. For both types of devices, the I - V curves in different cycles almost overlap with each other. Based on these results, the C2C variations (characterized by the DC voltage measurements) of the nonvolatile and volatile FDs are calculated to be ~6% and ~7%, respectively.

Figure R9. Multi-cycle I - V characteristics (15 cycles) for (a) nonvolatile and (b) volatile FDs. (c) Condensed plot of the LTP/LTD processes (15 cycles) of the nonvolatile FD. (d) Read current evolutions of the volatile FD under 30 repeated “1 0 1 0” pulse trains, where “1” represents a -2.5 V/2 ms pulse while “0” represents a 0 V pulse.

Figure R9(c) shows the multi-cycle LTP/LTD characteristics of the nonvolatile FD. It is seen that the LTP/LTD curves in different cycles largely overlap with each other, demonstrating that the multilevel conductance states are repeatable during the cyclic LTP/LTD pulse measurements. On the other hand, Figure R9(d) displays the read current evolutions of the volatile FD under 30 repeated “1 0 1 0” pulse trains. The read current responses in different cycles show little deviations. Based on these results, the C2C

variations (characterized by the pulse measurements) of the nonvolatile and volatile FDs are calculated to be $\sim 8\%$ and $\sim 6\%$, respectively.

(b) D2D variations of nonvolatile and volatile FDs

Figure R10. *I-V* characteristics for (a) 15 nonvolatile FDs and (b) 22 volatile FDs. (c) LTP/LTD processes measured from 15 nonvolatile FDs. (d) Read current evolutions of 22 volatile FDs under the same pulse train of “1 0 1 0”, where “1” represents a -2.5 V/2 ms pulse while “0” represents a 0 V pulse.

Figures R10(a) and (b) present the *I-V* curves for multiple devices of nonvolatile and volatile FDs, respectively. The *I-V* curves deviate rather greatly from each other. The D2D variations (characterized by the DC voltage measurements) of the nonvolatile and volatile FDs are calculated to be $\sim 56\%$ and $\sim 60\%$, respectively.

Figure R10(c) shows the LTP/LTD processes measured from 15 nonvolatile FDs. The LTP/LTD curves from different devices have relatively wide distributions. On the other hand, Figure R10(d) presents the read current evolutions of 22 volatile FDs under the same “1 0 1 0” pulse train. The read current responses from different devices show relatively large deviations. Based on these results, the D2D variations (characterized by the pulse measurements) of the nonvolatile and volatile FDs are calculated to be $\sim 40\%$ and $\sim 45\%$, respectively.

(b) Discussion on the impacts of C2C and D2D variations on the RC performance

It is seen from the above results that the C2C variations of both the nonvolatile and volatile FDs are rather small ($\leq \sim 8\%$). In particular, the small C2C variation of the volatile FD is the key for the accurate temporal signal transformation through the reservoir.

The D2D variations of both the volatile and nonvolatile FDs are, however, relatively large ($\sim 60\%$ or below). Nevertheless, the D2D variation of the volatile FD is indeed favorable to expand the reservoir size, which can help to improve the RC performance.

On the other hand, the D2D variation of the nonvolatile FD is not a big issue because of the inherent fault tolerance of the readout network. In addition, applying a write-and-verify method to the nonvolatile FD can ensure relatively precise weight programming despite the existences of C2C and D2D variations. In fact, for the nonvolatile FD, the good retention, as demonstrated in Figure 2 in the main text, is of critical importance for the RC performance.

The above results have been added into the revised main text and SI (please see the “Device reliability and power consumption” section in the revised main text and Figures S28 and S29 in the revised SI).

Comment (2): The second question concern the algorithm. Did the authors tried to look at other approaches such as e-prop (Bellec et al.). In this approach, the dynamic of spikes is used for a learning process. Whereas, reservoir computing is a quite old paradigm that doesn't scale to complex task.

Reply: Thank you for recommending the spike-based online learning algorithm called e-prop [14], which is energy-efficient and hardware-friendly. In e-prop, the synaptic weights are modified by a local rule for synaptic plasticity, i.e., the loss gradient is proportional to a sum of products of local eligibility traces and top-down learning signals over time steps. The eligibility traces are locally computed in a feedforward manner, while the learning signals (after approximation) capture only the errors that arise at the current time step. E-prop is thus an online learning method which does not require the propagation of gradient backward in time. In addition, e-prop achieves the performance comparable to that of the backpropagation through time (BPTT), which is the best-known method for training recurrent neural networks in machine learning. Therefore, e-prop is a promising approach to realize on-chip learning of recurrent networks of spiking neurons (RSNNs) on neuromorphic hardware.

Following the reviewer's suggestion, we have thought about the implementation of e-prop using memristors. As described above, e-prop mainly relies on two types of signals: eligibility traces and learning signals, both of which need to be appropriately implemented.

For the eligibility traces, as their main components show the spike history dependence and spontaneous decay, they may thus be implemented with volatile memristors (like our volatile FD). For the (approximate) learning signals, they are computed through a weighted sum of currently arising errors and may thus be implemented with nonvolatile memristors (like our nonvolatile FD).

Because our nonvolatile and volatile FDs have been demonstrated to be well functioning in the hardware system of reservoir computing, we therefore think that they can also be used for the hardware implementation of e-prop. Certainly, our above idea about the implementation of e-prop using memristors like our FDs is very preliminary. Further research on this interesting topic is warranted.

The above prospect has been added into the “Discussion” section in the revised main text, and the recommend paper on e-prop has been cited (see Ref. [61] in the main text).

Reviewer #3

Overall evaluation: In this manuscript, the authors used combination of Ferroelectric thin films (Volatile and nonvolatile ferroelectric diodes) to realize the reservoir computing devices. The manuscript is well written with clear storyline, and the all-ferroelectric RC system has demonstrated good working function. I recommend publication of this manuscript in Nat. Commun, after authors clarify below questions.

Reply: We thank the reviewer for the positive feedback and valuable comments, which are helpful to improve our manuscript. Our point-to-point responses are presented as follows.

Comment (1): The imprint from the P-V loop for 19 Pa film is not persuasive enough. As the imprint is related with the gap. I suggest the authors to use PFM to provide evidence of the imprint from phase or amplitude loops, as compared with the non-volatile 15 Pa films.

Reply: Following the reviewer's suggestion, we have performed PFM imaging and hysteresis loop measurements to study the imprint effect.

Figure R11. PFM (a,c) phase images after the box-in-box writing (outer: -6 V; inner: 6 V) and (b,d) amplitude and phase hysteresis loops for (a,b) 15 Pa and (c,d) 19 Pa BFO films.

Figures R11(a) and (c) show the PFM phase images after the box-in-box writing on the 15 Pa and 19 Pa BFO films, respectively. It is observed that the domains in both films can be reversibly switched, verifying their ferroelectricity. However, the domain configurations in the as-grown regions of the two films are different. In the 19 Pa film, most of the as-grown regions are occupied by downward domains, while there are only a

few tiny and irregularly-shaped upward domains. The dominance of downward domains may suggest that a downward E_{imp} exists in the 19 Pa film. By contrast, in the 15 Pa film, the upward domains in the as-grown regions become much bigger, and their total area is comparable to that of the downward domains. This suggests that the E_{imp} becomes negligible in the 15 Pa film.

Figures R11(b) and (d) present the PFM amplitude and phase hysteresis loops of the 15 Pa and 19 Pa BFO films, respectively. Both films exhibit butterfly-like amplitude loops and square phase loops with 180° switching. However, while the amplitude and phase loops of the 15 Pa film are rather symmetric with respect to 0 V, those of the 19 Pa film exhibit apparent negative voltage offsets. This suggests again that a downward E_{imp} exists in the 19 Pa film while it is absent in the 15 Pa film.

Note that the above PFM results are well consistent with our previous P - V loop results regarding imprint. The PFM results have therefore been added in the revised SI as the evidence for imprint (please see Figure S3 in the revised SI).

Comment (2): The authors explains the downward imprint electric field due to the surface accumulated oxygen vacancies according to the XPS data. However, the drifting in 19 Pa samples is too small to make the conclusion. In addition, there is no explanation of why the oxygen vacancies are near surface of 19 Pa film. Are these inhomogeneous oxygen vacancies distribution stable over time or temperature or external electric field?

Reply: As shown in the Figure R12, which is an enlarged view of Figure S13(b) in the revised SI, the peak measured at the surface (i.e., at the 0 nm depth) is shifted by ~0.16 eV relative to those measured in the bulk regions (i.e., at the ~35 and ~70 nm depths). This

peak shift is indeed not very small considering that the upper limit of the peak shift (i.e., the binding energy difference between the Fe^{2+} and Fe^{3+} $2p_{3/2}$ peaks) is only ~ 1.3 eV [16]. In addition, this peak shift is still greater than the energy resolution of our XPS system, i.e., 0.05 eV [17,18]. It may thus be safe to deduce that the Fe valence is lower at the surface than in the bulk regions. Further research is warranted to fully confirm the Fe valence variation and associated inhomogeneous oxygen vacancy distribution in the 19 Pa film.

Figure R12. XPS spectra of Fe $2p_{3/2}$ measured at different depths (0, ~ 35 , and ~ 70 nm) of the 19 Pa BFO film, with inset showing an enlarged view of the peaks.

In terms of the origin for the inhomogeneous oxygen vacancy distribution in the 19 Pa BFO film, we conjecture that during the growth of a BFO film, a sufficiently high oxygen pressure on the surface may cause the bulk oxygen vacancies to migrate to the surface (using the oxygen pressure gradient as the driving force) [19]. However, when the oxygen pressure on the surface is relatively low (e.g., 15 Pa), the bulk oxygen vacancies may still remain in the bulk region due to the insufficient oxygen pressure gradient. This

explanation has been added into the revised SI (please see the discussion on Figure S13 in the revised SI).

Figure R13. P - V loops of the Pt/BFO (19 Pa)/SRO device before and after applying a DC voltage sweep of $3\text{ V} \rightarrow -3\text{ V} \rightarrow 3\text{ V}$ to it.

The inhomogeneous oxygen vacancy distribution is stable against electric field (at room temperature), as demonstrated as follows. First, the P - V loop of the Pt/BFO (19 Pa)/SRO device measured with $\pm 4\text{ V}$ triangular pulses shows a negative voltage offset and a gap, suggesting that there is a downward E_{imp} . This E_{imp} should be persistent against the voltage pulse; otherwise, the P - V loop would not show such voltage offset and gap. Second, as shown in Figure R13, the P - V loop almost does not change after applying a DC voltage sweep of $3\text{ V} \rightarrow -3\text{ V} \rightarrow 3\text{ V}$ to the device, suggesting that the E_{imp} remains almost unchanged after applying the DC voltage sweep. The invariance of E_{imp} under both voltage pulse and DC voltage sweep implies that the inhomogeneous oxygen vacancy distribution is stable against both pulsed and DC electric fields. Note that this conclusion is valid throughout this study because the pulsed and DC voltages applied in this study are no larger than $\pm 4\text{ V}$ and $\pm 3\text{ V}$, respectively. Further enhancing the electric field may invalidate this conclusion, which is out of the scope of this study.

Figure R14. P - V loops of the Pt/BFO (19 Pa)/SRO device before and after leaving it in the ambient air at room temperature for 30 days.

We then turn to the stability of the inhomogeneous oxygen vacancy distribution over time. Figure R14 shows the initial P - V loop of the Pt/BFO (19 Pa)/SRO device and that measured after leaving the device in the ambient air at room temperature for 30 days. The two P - V loops almost overlap, suggesting that the E_{imp} almost does not change after a long duration. This in turn implies that the inhomogeneous oxygen vacancy distribution is stable over time (at room temperature).

Figure R15. P - V loops of the Pt/BFO (19 Pa)/SRO device before and after the thermal treatment, where the device was set to the P_{up} state and then annealed at 250 °C for 60 min in air.

The temperature stability of the inhomogeneous oxygen vacancy distribution was also investigated. The initial P - V loop of the Pt/BFO (19 Pa)/SRO device was first measured at room temperature. Then, the device was set to the P_{up} state and then annealed at 250 °C for 60 min in air. After this treatment, the P - V loop exhibits a positive voltage offset (see Figure R15), suggesting that the direction of E_{imp} is pointing upward. This in turn implies that the oxygen vacancies may migrate toward the bottom interface and accumulate there, causing the formation of an upward E_{imp} . In terms of the origin for the oxygen vacancy migration, we think that the oxygen vacancies become more mobile at high temperature [20], and they may migrate toward the bottom interface to compensate the negative polarization charge since the P_{up} state is established beforehand [21,22]. These results therefore suggest that the oxygen vacancy distribution can be changed at high temperature.

Nevertheless, because all the electrical measurements in this study were performed at room temperature, the inhomogeneous oxygen vacancy distribution and associated downward E_{imp} in the 19 Pa film can be considered as relatively stable.

The above results of Figures R13-R15 have been added into the revised SI (please see Figures S10-S12 in the revised SI).

Comment (3): What's the domain configuration (up and down domain ratio and distributions) like for 19Pa and 15Pa film? Does the local IV match the macroscopic resistive switching (on electrode), especially at up and down domain respectively.

Reply: The domain configurations in the 19 Pa and 15 Pa BFO films are shown in Figure R11. In the 19 Pa film [Figure R11(c)], most of the as-grown regions are occupied by downward domains. There are only a few tiny and irregularly-shaped upward domains

randomly distributed within the matrix of downward domains. The dominance of downward domains may suggest the existence of a downward E_{imp} in the 19 Pa film. By contrast, in the 15 Pa film [Figure R11(a)], the upward domains in the as-grown regions become much bigger, and they are randomly mixed with the downward domains. The total areas of the upward and downward domains are comparable, suggesting that negligible E_{imp} exists in the 15 Pa film.

We have tried measuring the local I - V characteristics in the different regions with upward and downward domains. Unfortunately, as shown in Figure R16, no conductive currents are observable even at 5 V (larger than the coercive voltages). This means that the conductive currents are too low, even below the current resolution of our CAFM system, disallowing any resistive switching behavior to be observed. The very low conductive current may be caused by two factors: a) the high contact barrier and b) the small contact area between the AFM tip and the bare BFO film.

Figure R16. Local I - V characteristics measured by locating the AFM tip in the regions with upward and downward domains.

References

1. Moon, J. et al. Temporal data classification and forecasting using a memristor-based reservoir computing system. *Nat. Electron.* **2**, 480-487 (2019).
2. Du, C. et al. Reservoir computing using dynamic memristors for temporal information processing. *Nat. Commun.* **8**, 2204 (2017).
3. Zhong, Y. et al. Dynamic memristor-based reservoir computing for high-efficiency temporal signal processing. *Nat. Commun.* **12**, 408 (2021).
4. Toprasertpong, K. et al. Reservoir computing on a silicon platform with a ferroelectric field-effect transistor. *Commun. Eng.* **1**, 21 (2022).
5. Pintilie, L. et al. Ferroelectric polarization-leakage current relation in high quality epitaxial Pb(Zr, Ti)O₃ films. *Phys. Rev. B* **75**, 104103 (2007).
6. Yu, J. et al. Energy efficient and robust reservoir computing system using ultrathin (3.5 nm) ferroelectric tunneling junctions for temporal data learning. In *2021 Symposium on VLSI Technology 1-2* (2021).
7. Tang, M. et al. A Compact Fully Ferroelectric-FETs Reservoir Computing Network With Sub-100 ns Operating Speed. *IEEE Electron Device Lett.* **9**, 43 (2022).
8. Liu, K. et al. Multilayer reservoir computing based on ferroelectric α -In₂Se₃ for hierarchical information processing. *Adv. Mater.* **22**, 2108826 (2022).
9. Liu, K. et al. An optoelectronic synapse based on α -In₂Se₃ with controllable temporal dynamics for multimode and multiscale reservoir computing. *Nat. Electron.* **22**, 1-13 (2022).
10. Choi, T. et al. Switchable ferroelectric diode and photovoltaic effect in BiFeO₃. *Science* **324**, 63-66 (2009).
11. Park, S. O. et al. Experimental demonstration of highly reliable dynamic memristor for artificial neuron and neuromorphic computing. *Nat. Commun.* **13**, 2888 (2022).

12. Midya, R. et al. Reservoir computing using diffusive memristors. *Adv. Intell. Syst.* **1**, 1900084 (2019).
13. Zhong, Y. et al. A memristor-based analogue reservoir computing system for real-time and power-efficient signal processing. *Nat. Electron.* **5**, 672-681 (2022).
14. Bellec, G. et al. A solution to the learning dilemma for recurrent networks of spiking neurons. *Nat. Commun.* **11**, 3625 (2020).
15. Milano, G. et al. In materia reservoir computing with a fully memristive architecture based on self-organizing nanowire networks. *Nat. Mater.* **21**, 195-202 (2022).
16. Kozakov, A. T. et al. X-ray photoelectron study of the valence state of iron in iron-containing single-crystal (BiFeO_3 , $\text{PbFe}_{1/2}\text{Nb}_{1/2}\text{O}_3$), and ceramic ($\text{BaFe}_{1/2}\text{Nb}_{1/2}\text{O}_3$) multiferroics. *J. Electron Spectrosc.* **184**, 16-23 (2011).
17. Li, C. et al. TiO_2 Coated polypropylene membrane by atomic layer deposition for oil-water mixture separation. *Adv. Fiber Mater.* **3**, 138-146 (2021).
18. Bai, K. et al. Selenium nanoparticles-loaded chitosan/citrate complex and its protection against oxidative stress in D-galactose-induced aging mice. *J. nanobiotechnology* **15**, 92 (2017).
19. Lu, Y. et al. Investigation of In-doped $\text{BaFeO}_{3-\delta}$ perovskite-type oxygen permeable membranes. *J. Mater. Chem. A.* **3**, 6202-6214 (2015).
20. Pike, G. E. et al. Voltage offsets in $(\text{Pb}, \text{La})(\text{Zr}, \text{Ti})\text{O}_3$ thin films. *Appl. Phys. Lett.* **66**, 484-486 (1995).
21. Lee, D. et al. Polarity control of carrier injection at ferroelectric/metal interfaces for electrically switchable diode and photovoltaic effects. *Phys. Rev. B* **84**, 125305 (2011).
22. Li, W. et al. Polarization-Dominated Internal Timing Mechanism in a Ferroelectric Second-Order Memristor. *Phys. Rev. Appl.* **19**, 014054 (2023).

REVIEWER COMMENTS

Reviewer #1 (Remarks to the Author):

The authors have made commendable efforts to answer the queries of all reviewers and put together a really nice response. The paper is much improved as a result, and I believe it does deserve publication in Nature Communications. I congratulate the authors for the high quality of their response.

However, I'm going to ask for a couple more things to be done. There has been somewhat of a flood of neuromorphic computing papers recently, and there is a variable quality amongst them especially in the 'behind the scenes' workings of the actual computation. This can be really difficult to understand for a non specialist, and being published in a journal such as nature comms is essentially a vote by the community that a certain paper has thought about things properly and used a good process. This revised paper is really close to that level of quality, but needs a bit more work on some of the benchmark testing.

I still think the curvature discrimination task is currently of a significantly lower standard than the rest of the paper. I think its clear its a task that the authors designed themselves, as its lacking the design of a good neuromorphic benchmark - which can be a challenging thing to do well.

The issues are in the size of the training and testing set. 2 previously unseen data points in a test set is not up to the standards of a journal such as nature comms. Also, as I guessed the linear resistors were able to predict 10/12 of curves - the pre/post processing is clearly doing a lot of the computing here - always a dangerous sign for a neuromorphic computing benchmark task. Its a good sign that the unseen curves worked better for the FDs, but 2 data points is nowhere near large enough to make a proper conclusion from (think of the random variation here, flipping a coin could easily get a 100% or 0% accuracy in a test set of 2), and definitely too small to reasonably call a test set. Due to the linear resistor case having a 10/12 accuracy and the short dataset, I am going to request some additional work here before publication.

The authors need to generate a considerably larger set. They can do this by mathematically parameterising their curves and varying the parameters many time, adding gaussian noise of various amplitude or ideally both. The training set should be more like 100-150 points minimum, and the test set more like 30-50. I would not make such a recommendation if there were experimental reasons why sets of such length cannot be processed, but the authors have demonstrated their nice ability to do large datasets like the full MNIST.

I would like control experiments for this larger test and train set, both the linear resistor that the authors have already demonstrated, and a second test replacing each FD with a simple RELUs (rectified linear unit). The RELU function has very limited nonlinearity and no memory, so if the task performance is dominated by the nice physical characteristics of the FDs as the authors claim, the authors should have no problem significantly outperforming the RELUs. I would like the results of the control experiments to be presented in the main figure, not in the supplementary so that readers can easily see an open and clear comparison.

I would also like to see the RELU test for the MNIST and Henon map. I believe the FDs will have no problem outperforming the RELUs for the Henon map, it will be interesting to see the MNIST case. For the RELU control, I would like the same number of masks/virtual nodes being used as for the FDs, so a 1:1 control experiment, simply replacing each FD with a RELU function.

I realise this will be some additional work, but as the authors have already coded up the linear case, adding a RELU function should be simple and quick. The paper will be much more useful to the community with full control experiments, which are often the difference between great neuromorphic papers and those of questionable community value. The papers which make an effort to do these comparisons tend to receive much higher citations, of benefit to the authors and the journal.

Where the authors discuss the various means of neuromorphic computing, it would be nice to include mention of recent developments using nanomagnetic arrays:

Dawidek, R. W., Hayward, T. J., Vidamour, I. T., Broomhall, T. J., Venkat, G., Mamoori, M. A., ... & Allwood, D. A. (2021). Dynamically driven emergence in a nanomagnetic system. *Advanced Functional Materials*, 31(15), 2008389.

Gartside, J. C., Stenning, K. D., Vanstone, A., Holder, H. H., Arroo, D. M., Dion, T., ... & Branford, W. R. (2022). Reconfigurable training and reservoir computing in an artificial spin-vortex ice via spin-wave fingerprinting. *Nature Nanotechnology*, 17(5), 460-469.

Allwood, D. A., Ellis, M. O., Griffin, D., Hayward, T. J., Manneschi, L., Musameh, M. F. K., ... & Wringe, C. (2023). A perspective on physical reservoir computing with nanomagnetic devices. *Applied Physics Letters*, 122(4), 040501.

Reviewer #3 (Remarks to the Author):

The authors have adequately addressed all my questions. I recommend the publication of this manuscript on Nature Communications.

Responses to Reviewers' Comments

Reviewer #1

Overall evaluation: The authors have made commendable efforts to answer the queries of all reviewers and put together a really nice response. The paper is much improved as a result, and I believe it does deserve publication in Nature Communications. I congratulate the authors for the high quality of their response.

However, I'm going to ask for a couple more things to be done. There has been somewhat of a flood of neuromorphic computing papers recently, and there is a variable quality amongst them especially in the 'behind the scenes' workings of the actual computation. This can be really difficult to understand for a non specialist, and being published in a journal such as nature comms is essentially a vote by the community that a certain paper has thought about things properly and used a good process. This revised paper is really close to that level of quality, but needs a bit more work on some of the benchmark testing.

Reply: We would like to thank the reviewer for the positive feedback to our previous responses and revisions. We also appreciate the reviewer very much for raising further comments which are helpful to improve the manuscript. Our responses to the specific comments are presented below.

Comment (1): I still think the curvature discrimination task is currently of a significantly lower standard than the rest of the paper. I think it's clear it's a task that the authors

designed themselves, as it's lacking the design of a good neuromorphic benchmark - which can be a challenging thing to do well.

The issues are in the size of the training and testing set. 2 previously unseen data points in a test set is not up to the standards of a journal such as nature comms. Also, as I guessed the linear resistors were able to predict 10/12 of curves - the pre/post processing is clearly doing a lot of the computing here - always a dangerous sign for a neuromorphic computing benchmark task. It's a good sign that the unseen curves worked better for the FDs, but 2 data points is nowhere near large enough to make a proper conclusion from (think of the random variation here, flipping a coin could easily get a 100% or 0% accuracy in a test set of 2), and definitely too small to reasonably call a test set. Due to the linear resistor case having a 10/12 accuracy and the short dataset, I am going to request some additional work here before publication.

The authors need to generate a considerably larger set. They can do this by mathematically parameterising their curves and varying the parameters many time, adding gaussian noise of various amplitude or ideally both. The training set should be more like 100-150 points minimum, and the test set more like 30-50. I would not make such a recommendation if there were experimental reasons why sets of such length cannot be processed, but the authors have demonstrated their nice ability to do large datasets like the full MNIST.

I would like control experiments for this larger test and train set, both the linear resistor that the authors have already demonstrated, and a second test replacing each FD with a simple RELUs (rectified linear unit). The RELU function has very limited nonlinearity and no memory, so if the task performance is dominated by the nice physical

characteristics of the FDs as the authors claim, the authors should have no problem significantly outperforming the RELUs. I would like the results of the control experiments to be presented in the main figure, not in the supplementary so that readers can easily see an open and clear comparison.

Reply: Thanks for this good suggestion. We have parameterized the designed curves, and increased the sizes of the training and test sets.

The original curve follows the equation:

$$y = ax^2 \tag{R1}$$

where a is a parameter which can be varied. The curve is then rotated around the origin by an angle of θ (θ is a variable parameter). Afterward, the curve is shifted horizontally and vertically by p and q , respective, where p and q are also variable parameters. The horizontal coordinate interval of the curve is set to be $[-4, 4]$. By varying the parameters a , θ , p , and q (see Table R1), 138 different curves are generated. 102 curves are then selected from the whole 138 curves and constitute the training set [see Figure R2(a)], while the rest 36 curves constitute the test set [see Figure R2(b)].

Table R1. Parameters of the curves used in the curvature discrimination task. The first 102 curves constitute the training set, while the test set is composed of the rest 36 curves.

Curve number	a	θ (degree)	p	q
1	0.8	-30	3.6	3.45
2	0.8	-30	3.6	3.5
3	2.5	-42	4	3.5
4	1	-30	3.6	3.5
5	0.5	-20	3.2	3.5
6	0.6	-20	3.1	3.5
7	0.45	-0.5	0	3.55

8	0.4	-0.5	0	3.55
9	0.4	-5	0.6	3.55
10	0.4	-5	0.8	3.55
11	0.3	-5	0.5	3.55
12	0.5	-20	3	3.4
13	0.4	-22	3.1	3.4
14	0.45	-22	2.9	3.4
15	0.4	-16	2.5	3.5
16	0.35	-16	2.5	3.5
17	0.35	-12	2.2	3.5
18	0.3	-14	2	3.5
19	0.4	-5	1.2	3.5
20	0.35	-5	1.2	3.5
21	0.35	-5	1.6	3.5
22	0.35	-10	1.6	3.5
23	0.28	-13	1.6	3.5
24	0.28	13	-1.6	3.5
25	0.35	10	-1.6	3.5
26	0.35	5	-1.6	3.5
27	0.35	5	-1.2	3.5
28	0.4	5	-1.2	3.5
29	0.3	14	-2	3.5
30	0.35	12	-2.2	3.5
31	0.35	16	-2.5	3.5
32	0.4	16	-2.5	3.5
33	0.45	22	-2.9	3.4
34	0.4	22	-3.1	3.4
35	0.5	20	-3	3.4
36	0.3	5	-0.5	3.55
37	0.4	5	-0.8	3.55
38	0.4	5	-0.6	3.55
39	0.4	0.5	0	3.55
40	0.45	0.5	0	3.55
41	0.6	20	-3.1	3.5
42	0.5	20	-3.2	3.5
43	1	30	-3.6	3.5
44	1	30	-3.8	3.5
45	1.3	30	-3.8	3.5
46	2.5	42	-4	3.5
47	2.5	38	-4	3.5
48	3.2	36	-3.9	3.5
49	3	36	-3.9	3.5
50	0.8	30	-3.6	3.5
51	0.8	30	-3.6	3.45
52	0.8	-150	3.6	-3.45
53	0.8	-150	3.6	-3.5
54	2.5	-138	4	-3.5
55	1	-150	3.6	-3.5
56	0.5	-160	3.2	-3.5
57	0.6	-160	3.1	-3.5
58	0.45	-179.5	0	-3.55

59	0.4	-179.5	0	-3.55
60	0.4	-175	0.6	-3.55
61	0.4	-175	0.8	-3.55
62	0.3	-175	0.5	-3.55
63	0.5	-160	3	-3.4
64	0.4	-158	3.1	-3.4
65	0.45	-158	2.9	-3.4
66	0.4	-164	2.5	-3.5
67	0.35	-164	2.5	-3.5
68	0.35	-168	2.2	-3.5
69	0.3	-166	2	-3.5
70	0.4	-175	1.2	-3.5
71	0.35	-175	1.2	-3.5
72	0.35	-175	1.6	-3.5
73	0.35	-170	1.6	-3.5
74	0.28	-167	1.6	-3.5
75	0.28	167	-1.6	-3.5
76	0.35	170	-1.6	-3.5
77	0.35	175	-1.6	-3.5
78	0.35	175	-1.2	-3.5
79	0.4	175	-1.2	-3.5
80	0.3	166	-2	-3.5
81	0.35	168	-2.2	-3.5
82	0.35	164	-2.5	-3.5
83	0.4	164	-2.5	-3.5
84	0.45	158	-2.9	-3.4
85	0.4	158	-3.1	-3.4
86	0.5	160	-3	-3.4
87	0.3	175	-0.5	-3.55
88	0.4	175	-0.8	-3.55
89	0.4	175	-0.6	-3.55
90	0.4	179.5	0	-3.55
91	0.45	179.5	0	-3.55
92	0.6	160	-3.1	-3.5
93	0.5	160	-3.2	-3.5
94	1	150	-3.6	-3.5
95	1	150	-3.8	-3.5
96	1.3	150	-3.8	-3.5
97	2.5	138	-4	-3.5
98	2.5	142	-4	-3.5
99	3.2	144	-3.9	-3.5
100	3	144	-3.9	-3.5
101	0.8	150	-3.6	-3.5
102	0.8	150	-3.6	-3.45
103	1	-30	3.6	3.45
104	3	-36	3.9	3.5
105	3.2	-36	3.9	3.5
106	0.4	-5	1.1	3.55
107	0.45	-20	2.9	3.4
108	0.35	-14	2.2	3.5
109	0.4	-10	1.2	3.5

110	0.35	-10	1.2	3.5
111	0.26	-10	1.4	3.55
112	1	30	-3.6	3.45
113	0.4	5	-1.1	3.55
114	0.45	20	-2.9	3.4
115	0.35	14	-2.2	3.5
116	0.4	10	-1.2	3.5
117	0.35	10	-1.2	3.5
118	0.26	10	-1.4	3.55
119	0.42	0	0	3.55
120	2.5	-34	3.87	3.55
121	1	-150	3.6	-3.45
122	3	-144	3.9	-3.5
123	3.2	-144	3.9	-3.5
124	0.4	-175	1.1	-3.55
125	0.45	-160	2.9	-3.4
126	0.35	-166	2.2	-3.5
127	0.4	-170	1.2	-3.5
128	0.35	-170	1.2	-3.5
129	0.26	-170	1.4	-3.55
130	1	150	-3.6	-3.45
131	0.4	175	-1.1	-3.55
132	0.45	160	-2.9	-3.4
133	0.35	166	-2.2	-3.5
134	0.4	170	-1.2	-3.5
135	0.35	170	-1.2	-3.5
136	0.26	170	-1.4	-3.55
137	0.42	-180	0	-3.55
138	2.5	-146	3.87	-3.55

Figure R1. Plots of the curves in the (a) training set and (b) test set.

The pre-processing of the curves, construction of the all-ferroelectric RC system, and training method are the same as those used previously (please refer to Pages 16 and 17 of the main text for details). As shown in Figure R2, the trained all-ferroelectric RC system achieves 100% accuracy on the test set.

Figure R2. Comparison of the test accuracies for curvature discrimination achieved by the all-ferroelectric RC system with a volatile FD-based reservoir and two control RC systems with linear resistor- and ReLU-based reservoirs.

For the performance comparison, we have designed and tested two control RC systems, where the volatile FDs in the reservoir are replaced with linear resistors and ReLU functions, respectively [see Figure R3(a)]. The ReLU function is expressed by

$$f(x) = \max(t, x), \quad (\text{R2})$$

where the input x is the y -coordinate of the point on the input curve, and t is a parameter which can be optimized. Note that for different tasks, x and t can be changed accordingly.

As shown in Figure R2, both the linear resistor- and ReLU-based RC systems achieve the same accuracy of 83.3% on the test set. The same accuracy of the two control systems

suggests that the nonlinearity of the ReLU function does not contribute to the RC performance in this simple task.

Figure R2 also presents that the accuracies of the two control systems are apparently lower than the 100% accuracy of the volatile FD-based RC system (i.e., the all-ferroelectric RC system). To understand why the all-ferroelectric RC system outperforms the two control systems, typical misclassified results obtained from the two control systems are specifically analyzed. Figure R3(b) displays two typical curves from the test set which are misclassified by the two control systems. When inputting the curve with negative (positive) curvature, both the linear resistor and ReLU function corresponding to the middle section of the curve produce a lower (higher) output, as shown in Figure R3(c). This is the main cause for the two control systems to make wrong classification [Figure R3(d)] given the readout weights trained offline. Why the linear resistor and ReLU function produce such outputs is simply because they have no memory effect and hence the relative height of the last point in the middle section of the curve determines the magnitude of the output.

By contrast, the volatile FD produces different current outputs [see Figure R3(c)], which is well attributed to its memory effect allowing the pulse history to influence the conductance. This in turn results in a correct classification given the readout weights trained offline, as shown in Figure R3(d).

Figure R3. Control experiments of curvature discrimination. (a) Schematics of two control RC systems where the volatile FDs in the reservoir are replaced by linear resistors and ReLU functions, respectively. (b) Two typical curves from the test set which are misclassified by the two control RC systems. (c) Outputs of the volatile FDs, linear resistors, and ReLU functions when inputting the curves in b. Device 1-3 correspond to beginning, middle, and end sections of the curve, respectively. (d) Compared results on the discrimination of the curves in b, obtained from the RC systems based on the volatile FDs, linear resistors, and ReLU functions.

The above compared results therefore confirm that the volatile FD-based reservoir is the key for the superior performance of our RC system in the curvature discrimination.

Following the reviewer's request to present the results of control experiments in the main text, we have placed Figure R2 into the main text (see Figure 4f). On the other hand, due to the limited space of the figures in the main text, Figure R1 and R3 have been placed in the SI (see Figure S20 and S21, respectively).

Comment (2): I would also like to see the RELU test for the MNIST and Henon map. I believe the FDs will have no problem outperforming the RELUs for the Henon map, it will be interesting to see the MNIST case. For the RELU control, I would like the same number of masks/virtual nodes being used as for the FDs, so a 1:1 control experiment, simply replacing each FD with a RELU function.

I realise this will be some additional work, but as the authors have already coded up the linear case, adding a RELU function should be simple and quick. The paper will be much more useful to the community with full control experiments, which are often the difference between great neuromorphic papers and those of questionable community value. The papers which make an effort to do these comparisons tend to receive much higher citations, of benefit to the authors and the journal.

Reply: Following the reviewer's valuable suggestion, we have performed the control experiments using the ReLU function for the MNIST handwritten digit recognition and Hénon map prediction. In these control experiments, only the volatile FDs in the reservoir are replaced by linear resistors or ReLU functions, while the other factors like the mask process and readout network are all unchanged.

The compared results on the MNIST handwritten digit recognition are shown in Figure R4(b). It is seen that the linear resistor-based RC system achieves 84.3% accuracy on the test set, while the ReLU-based RC system achieves a higher accuracy of 86.3%. The higher performance of the ReLU-based RC system may be attributed to the nonlinearity of the ReLU function. However, the accuracies of both the two control systems are lower than the accuracy of 89.5% achieved by the volatile FD-based RC system (i.e., the all-ferroelectric RC system). It is deducible that the combined memory effect and nonlinearity of the volatile FD lead to the high performance of the volatile FD-based RC system.

Figure R4. Control experiments of MNIST handwritten digit recognition. (a) Schematics of two control RC systems where the volatile FDs in the reservoir are replaced by linear resistors and ReLU functions, respectively. (b) Comparison of accuracies on the test set achieved by the volatile FD-, linear resistor-, and ReLU-based RC systems.

The Hénon map time series predicted by the linear resistor- [Figure 5(a)] and ReLU-based [Figure 5(c)] RC systems are shown in Figure 5(b) and (d), respectively. The NRMSE values on the test set achieved by the linear resistor- and ReLU-based RC systems are 0.70 and 0.27, respectively. The lower NRMSE value of the ReLU-based RC system may result from the nonlinearity of the ReLU function. However, the NRMSE values of

both the two control systems are much higher than that achieved by the volatile FD-based RC system, i.e., 0.017. Such superior performance of the volatile FD-based RC system is well attributed to the combined memory effect and nonlinearity of the volatile FD.

Figure R5. Control experiments of Hénon map prediction. (a) Schematic of a control RC system where the volatile FDs in the reservoir are replaced by linear resistors. (b) Predicted time series versus ideal targets obtained from the RC system shown in b. (c) Schematic of a control RC system where the volatile FDs in the reservoir are replaced by ReLU functions. (d) Predicted time series versus ideal targets obtained from the RC system shown in c.

The above results have been added into the revised SI. Please see Figure S26 and S28, and their related discussion.

Comment (3): Where the authors discuss the various means of neuromorphic computing, it would be nice to include mention of recent developments using nanomagnetic arrays:

Dawidek, R. W., Hayward, T. J., Vidamour, I. T., Broomhall, T. J., Venkat, G., Mamoori, M. A., ... & Allwood, D. A. (2021). Dynamically driven emergence in a nanomagnetic system. *Advanced Functional Materials*, 31(15), 2008389.

Gartside, J. C., Stenning, K. D., Vanstone, A., Holder, H. H., Arroo, D. M., Dion, T., ... & Branford, W. R. (2022). Reconfigurable training and reservoir computing in an artificial spin-vortex ice via spin-wave fingerprinting. *Nature Nanotechnology*, 17(5), 460-469.

Allwood, D. A., Ellis, M. O., Griffin, D., Hayward, T. J., Manneschi, L., Musameh, M. F. K., ... & Wringe, C. (2023). A perspective on physical reservoir computing with nanomagnetic devices. *Applied Physics Letters*, 122(4), 040501.

Reply: We thank the reviewer for recommending the recent papers on the neuromorphic computing based on nanomagnetic arrays.

The first paper reports the development of arrays of interconnected magnetic ring-shaped nanowires, which exhibit a dynamic, field-dependent domain wall population under an applied rotating magnetic field. This emergent behavior allows the magnetic ring array to implement the RC, and the simulated results show that the magnetic ring array-based RC system achieves near 100% accuracy in classifying spoken digits for single speakers.

The second paper demonstrates an artificial spin-vortex ice (ASVI) array that can host both Ising-like macrospins and vortices. The ASVI array exhibits history-dependent conversion between vortex and macrospin as well as fading memory behavior under global magnetic field cycles, as characterized by ferromagnetic resonance measurements. With

these properties, the ASVI array is capable of being used for RC, as demonstrated with the tasks of nonlinear mapping transformation and chaotic time-series forecasting.

The third paper presents a comprehensive review on the nanomagnetic materials and devices for RC, including spin torque oscillators, spin ice arrays, skyrmion textures, superparamagnetic arrays, magnonic systems, and domain wall devices. In particular, the physical properties and mechanisms, training methods, simulation tools, and characterization methods of various nanomagnetic systems for RC, as well as the challenges in this field, are discussed in detail.

These three papers have been mentioned in the Introduction part of the revised manuscript:

“Most previous studies have focused on the hardware implementation of the reservoir by using (volatile) diffusive memristors^{6,7,10,14-22}, **nanomagnetic systems²³ (including spintronic oscillators²⁴, magnetic nanorings²⁵, spin ices²⁶, and magnonic systems²⁷)**, self-organized nano-networks^{9,28}, electrochemical transistors^{8,12,29}, and so on.”

However, to maintain the flow of our paper and also due to the length issue, we have not given a detailed introduction to these nanomagnetic systems.

REVIEWERS' COMMENTS

Reviewer #1 (Remarks to the Author):

The authors have done a great job addressing all requests and the manuscript is now much improved. Thanks for addressing all the comments. I know fully recommend this paper for publication in Nature Communications.

One small point - the fact that the ReLU and linear control tests get exactly the same accuracy for the curve discrimination in figure R2 (83.3%) shows that the ReLU was actually a poor choice here for providing a non-linear control. ReLU is only nonlinear if some of its inputs are positive and some are negative, it looks here like it only got passed positive values, or else there was an issue in the code used to implement it. Numerically identical values show that it must be the same as the linear case.

While the ReLU worked really nicely as a control for the other tasks, here it doesn't really make sense. I'd invite the authors to potentially try adding an activation function that is nonlinear even for all-positive values, eg tanh or sigmoid, but I know I've made a lot of requests already and the authors have generously agreed to do them. I would say this is up to the authors/editors - the benefit to the paper and authors will be that if readers see the existing identical accuracy between linear and ReLU, they may wonder if a mistake has been made in implementation or if it was an odd choice of control function.

Other than that (which again, is an entirely optional addition) the paper is great and ready to publish.

Responses to Reviewers' Comments

Reviewer #1

Overall evaluation: The authors have done a great job addressing all requests and the manuscript is now much improved. Thanks for addressing all the comments. I know fully recommend this paper for publication in Nature Communications.

Reply: We are happy that the reviewer was satisfied with our previous responses and recommended our manuscript for publication in Nature Communications. We also appreciate the reviewer very much for providing valuable comments which are helpful to improve the manuscript.

Comment (1): One small point - the fact that the ReLU and linear control tests get exactly the same accuracy for the curve discrimination in figure R2 (83.3%) shows that the ReLU was actually a poor choice here for providing a non-linear control. ReLU is only nonlinear if some of its inputs are positive and some are negative, it looks here like it only got passed positive values, or else there was an issue in the code used to implement it. Numerically identical values show that it must be the same as the linear case.

While the ReLU worked really nicely as a control for the other tasks, here it doesn't really make sense. I'd invite the authors to potentially try adding an activation function that is nonlinear even for all-positive values, eg tanh or sigmoid, but I know I've made a lot of requests already and the authors have generously agreed to do them. I would say this is up to the authors/editors - the benefit to the paper and authors will be that if readers see the

existing identical accuracy between linear and ReLU, they may wonder if a mistake has been made in implementation or if it was an odd choice of control function.

Other than that (which again, is an entirely optional addition) the paper is great and ready to publish.

Reply: Thanks for this good suggestion. The ReLU function used here is expressed by

$$f(x) = \max(t, x), \quad (\text{R1})$$

where the input x is the y -coordinate of the point on the input curve, and t is a parameter which can be optimized.

As seen from the above definition, some of the inputs can be positive and some can be negative. This is indeed the case where the ReLU function is applicable. In addition, even though the inputs are all positive, the ReLU function used here can still provide nonlinearity if the parameter t adopts a positive value. This is different from the conventional ReLU function where t is always 0 (note: because t is always 0, the conventional ReLU function cannot provide nonlinearity for all-positive values, as pointed out by the reviewer).

Although the ReLU function used here can provide nonlinearity for all-positive values, the ReLU-based RC system fails to achieve a higher accuracy than the linear resistor-based one. The same accuracy achieved by these two RC systems may be attributed to the fact that both the ReLU function and linear resistor have no memory effect, and the memory effect is more critical than the nonlinearity for the RC performance in this simple task. The detailed analyses can be found in the Supplementary Fig. S21 and Note 2 of the previous version of the manuscript.

Although the same accuracy achieved by the ReLU- and linear resistor-based RC systems in the curvature discrimination task can be explained, we have still replaced the ReLU function with the sigmoid function following the reviewer's suggestion. The results are shown in Fig. 4f in the main text and Supplementary Fig. S26 and S28 of the revised manuscript. It is interesting to note that the sigmoid-based RC system also exhibits the same accuracy as the linear resistor-based one in the curvature discrimination task. This can also be explained by the absence of the memory effect in the sigmoid function. In other tasks, the sigmoid-based RC system achieves higher performance than the linear resistor-based one, but its performance is lower than that of the all-ferroelectric RC system. The detailed analyses can be found in Supplementary Fig. S26, Fig. S28, Note 2, and Note 3 of the revised manuscript.